# Sensitivities of the MJO Forecasts on Configurations of Physics in the ECMWF Global Model

Jun–Ichi Yano[1] and Nils P. Wedi[2]

[1]CNRM, UMR 3589 (CNRS), Météo-France, 31057 Toulouse Cedex, France
[2]European Center for Medium–Range Weather Forecast, Reading, UK

**Correspondence:** Jun–Ichi Yano (jiy.gfder@gmail.com)

**Abstract.** Sensitivities of MJO forecasts on various different configurations of the parametrised physics are examined with the global model of ECMWF's Integrated Forecasting System (IFS). The motivation of the study has been to simulate the MJO as a nonlinear free wave under active interactions with higher–latitude Rossby waves. To emulate free dynamics in IFS, various momentum dissipation terms ("friction") as well as diabatic heating are selectively turned off over the tropics for the range of the latitudes 20S-20N. The reduction of friction sometimes improves the MJO forecasts, but without any systematic tendency. Contrary to the original motivation, emulating a free dynamics with an operational forecast model turns out to be rather difficult, because forecast performance sensitively depends on the specific type of friction turned off. The result suggests a need for theoretical investigations that much more closely follow the actual formulations of model physics: a naive approach with a dichotomy of with or without friction simply fails to elucidate the rich behaviour of complex operational models. The paper further exposes the importance of physical processes other than convection for simulating the MJO in global forecast models.

## 1 Introduction

The Madden-Julian oscillation (MJO: Zhang 2005) is a prominent tropical variability that many global atmospheric models still have difficulties in simulating. In the case of the ECMWF Integrated Forecasting System (IFS), the forecast of the propagation of a pre–existing MJO has much improved in recent years (Vitart 2014), typically providing persistent MJO signals well beyond the medium-range forecast. However, the IFS still suffers from some difficulties, especially in predicting the onset of MJOs. Needs for a capacity of extended MJO forecasts are becoming more important with increasing demand for extended forecasts up to a subseasonal range (3–4 weeks) and because the MJO is one of the most prominent and persistent tropical signals to be forecast over this time scale (*cf.*, Kim *et al.* 2018).

From an operational point of view, the MJO is typically considered physically forced in the sense that the physical parametrisation (or 'physics' for short) in the models are the key for improving the simulation of the MJO, rather than a problem of the dynamical core (*e.g.*, Hirons *et al.* 2013a, b). The most crucial physical process to be considered is deep convection, that is typically parametrised as a subgrid-scale process in global models (Plant and Yano 2015). A majority of the existing theories for the MJO are based on a certain coupling of the large–scale dynamics with convection (*e.g.*, Hayashi 1970, Lindzen 1974,

Emanuel 1987, Yano and Emanuel 1991, Majda and Stechmann 2009, Fuchs and Raymond 2017: see also reviews by Zhang *et al.* 2020, Jiang *et al.* 2020). For this reason, a general expectation is that simulations and forecasts of the MJO in the global models must be improved by improving the parametrization of deep convection (*cf.*, Jiang *et al.* 2015, 2020) as well as shallow convection (*cf.*, Pilon *et al.* 2015). For this reason, existing sensitivity studies on MJO simulations almost exclusively focus on convection parameterizations (*e.g.*, Hirons *et al.* 2013a, b, Pilon *et al.* 2015).

The present study examines the sensitivity of the MJO forecasts on physics from a different perspective proposed by Yano and Bonazzola (2009), Yano *et al.* (2009), Wedi and Smolarkiewicz (2010), Yano and Tribbia (2017), Rostam and Zeitlin (2019), and Wang *et al.* (2019). According to their perspective, the tropical large–scale dynamics in general and the MJO specifically can be understood in terms of *free* Rossby–wave dynamics, in which model "physics" may still play a role, but secondary to the initiation and evolution. More specifically, Yano and Tribbia (2017), and Rostam and Zeitlin (2019) propose

that the MJO is basically understood in terms of a dipolar vortex (vortex pair) symmetric to the equator, described by a nonlinear analytical solution, called *modon*, which propagates eastwards as observed for the MJO. To investigate this possibility of the MJO as a free dynamics in the context of the operational global forecasts, we take the ECMWF global model (IFS) as a basic framework, and perform extensive physical sensitivity experiments. See Sec. 2.1 for model details.

To emulate a free dynamics within IFS, physical tendencies of some variables are selectively turned off so that the resulting

sensitivities to the corresponding MJO forecasts can be examined. A key process to be turned off to emulate a free dynamics is the surface friction, or momentum dissipation more generally. This process has been expected to potentially play a crucial role in the MJO dynamics. A classical work by Chang (1977) makes this point by invoking the surface friction as a mechanism to slow down the propagation speed of the eastward-propagating free Kelvin wave to a degree comparable to that of the MJO. The frictional wave-CISK theories by Wang (1988) and Salby *et al.* (1994) also invoke frictional moisture convergence as a

key ingredient in addition to deep convection for explaining the basic dynamics of the MJO. Along with the surface friction, diabatic heating is another key process to be turned off for achieving a free dynamics.

A shortcoming of the free–wave theory of the MJO is that it does not explain by itself an MJO initiation. Thus, when physical forcings are turned off from a model, an initiation mechanism must be sought. Particular attention is paid to the potential importance of interactions of the MJO with higher-latitude dynamics for this reason. Weickmann *et al.* (1985), Knutson and

50 Weickmann (1987) suggest that the interactions with Rossby-wave trains from and to higher latitudes are intrinsic parts of the MJO dynamics. Hsu *et al.* (1990), Gustafson and Weare (2004), Ray and Zhang (2010), Ray and Li (2013), Zhao *et al.* (2013), and Wang *et al.* (2019) further suggest that Rossby-wave trains from the northern-hemisphere higher-latitudes initiate MJOs. General importance of higher–latitude variability in MJO dynamics is also suggested by modelling of MJOs under an equatorial channel configuration, in which a properly prescribed lateral boundary condition is crucial (*cf.*, Hall *et al.* 2017 and references

therein).

For investigating these aspects of the MJO dynamics, we attempt to simulate the higher-latitude dynamics as properly as possible. In the following sensitivity experiments, a weighting of $\cos^6 \phi$ with $\phi$ the latitude is adopted so that the effects of the applied sensitivity rapidly tail off polewards of *ca.*, $20°$. Hence, when a certain process is turned off over the tropics, the tendency due to this process is multiplied by $1 - \cos^6 \phi$.

**Table 1.** Four major categories of experiments

| Category | Description |
| --- | --- |
| 1 | Control operational forecasts |
| 2 | OFF selected physical tendencies for the momentum (*e.g.*, shallow and deep convection, vertical eddy diffusion) |
| 3 | OFF physical tendency for the temperature (entropy) (due to shallow and deep convection, radiation and cloud phase changes) |
| 4 | OFF all physical tendencies as above for both momentum and temperature |

Under this general strategy, four major categories of experiments are performed as listed in Table 1. These experiments are designed to address the following questions: 1) Can the propagation of the MJO be simulated in a complex forecast model even if the diabatic heating due to convection is turned off? 2) To what extent can the simulated MJO be interpreted in terms of free Rossby–wave dynamics?

  To address the question 1), we turn off all the diabatic heating in the heat equation (entropy budget) so that an adiabatic
free dynamics regime is realised over the tropics. Here, it is crucial to turn off all the diabatic heating, because if the latent heating is turned off, but the radiative cooling tendency of the tropics is maintained, a steady state can only be maintained by turning the mean ascent (associated with moist convection) to a mean descent, that induces diabatic heating that balances the radiative cooling. We turn off the total diabatic heating so that the tendency for generating any vertical motion is suppressed, and a purely horizontal, quasi-nondivergent flow is realised.

To address the question 2), we turn off the non-conservative processes (*i.e.*, frictional dissipation in general) in the horizontal momentum equation, because we expect that the free Rossby–wave dynamics associated with the MJO are enhanced by turning off the momentum dissipation. As a result, we also expect that Rossby-wave interactions between the tropics and the higher latitudes are enhanced. A claim of MJO as a free Rossby wave also contains another important general implication that the MJO can be principally understood in terms of nondivergent, rotational flows. Thus, an important question to be investigated
is to what extent a non-divergent (rotational) component of the MJO is still maintained by selectively turning off the physics.

  An exploratory nature of the present investigation is emphasised. Unfortunately, our goal of emulating the free dynamics is not achieved in any obvious manner, without any systematically–identifiable trait in these sensitivity experiments. For example, the reduction of momentum–dissipation effects ("frictions") in the model does not lead to a simple improvement or deterioration of the MJO forecast. The paper focuses on elucidating these complex sensitivities of the MJO forecasts on different
configurations of the physics. Detailed descriptions of the results are presented as objectively as possible, with the purpose of elucidating real operational issues in improving the MJO forecasts. That is where theoretical investigations are strongly needed to better understand the model behaviour.

  For example, the role of friction in the MJO dynamics remains a key question since a pioneering study by Chang (1977). However, the majority of theoretical studies treat it simply as a Rayleigh friction (*cf.*, Sec. 4 of Yano *et al.* 2013 as a review
of this line of theoretical studies). The present study, in turn, shows that the actual contribution of friction in an operational

model is far more complex. Thus, a more serious effort to fill a gap between those idealised theoretical studies and operational problems is required.

The present study is unique in modelling studies by examining the roles of more specific physical processes in the MJO dynamics: for example, instead of turning off the whole momentum–dissipation process, individual momentum–dissipation processes are turned off one by one. This is in contrast to the mechanism–denial studies (*e.g.*, Kim *et al.* 2011, Ma and Kuang 2016), in which a whole process (*e.g.*, momentum dissipation, surface–flux evaporation) is typically turned off (see also *e.g.*, Crueger and Stevens 2015). The present study is also conceptually different from the mechanism–denial studies. The latter replaces the turned–off processes by climatologies, whereas the present study turns off a given process with a goal of getting closer to an idealized free dynamics.

However, there is a subtlety in turning off certain physics in a given model, because of their impact on the mean state and the nonlinearity of the system leading to various chain reactions and compensating behaviour with corresponding changes to MJO forecast skill. We find that a change of the results by turning off different physics hardly constitute simple additive processes. Previous studies have found significant changes in the energy cascade behaviour of the IFS model, controlled by certain physics or specific parts thereof (Malardel and Wedi 2016). A change of the tropical processes clearly influences the interactions of the tropical processes with those in higher latitudes. Subtle balances between higher latitudes and the tropics must therefore carefully be taken into account for a full interpretation of these sensitivity results.

The main contribution of the present study is to suggest that the MJO dynamics is not just a matter of its coupling with convection, but other physical processes, including friction, actively contributing in defining its dynamics. Another important, rather unintuitive result is a strong sensitivity of the MJO forecast on initial conditions. The following analysis is focused over the region of the Indian Ocean to the Western Pacific (90–180E), where main activities of the MJO are identified. Although the original study by Madden and Julian (1972) identifies the MJO as a global mode, as the analysis by Milliff and Madden (1996) shows, the continuous mode propagating eastwards beyond the Date Line is rather identified as a free Kelvin wave.

The next section describes the model used in this study (Sec. 2.1), the forecast cases (Sec. 2.2 and Sec. 2.3) and the analysis procedure (Sec. 2.4). Results are presented in Sec. 3 and the paper concludes with a discussion in Sec. 4.

## 2    Model, Forecast, Cases and Analysis Procedure

### 2.1    Model description

The IFS version cycle 43r3 (operational during 11 July 2017 - 5 June 2018) is used for the forecast experiments with TCo639 (average grid spacing 18 km) and with 137 vertical levels. IFS is a spectral transform model solving part of the solution in spectral space, where prognostic variables are represented by spherical harmonics. To calculate nonlinear terms in the equations of motion, to perform the nonlinear (semi-Lagrangian) advection, and to calculate the contributions of all physics schemes in grid point columns, the model fields are transformed into a representation in grid-point space. A cubic octahedral (reduced) Gaussian grid is used for this purpose, denoted by 'TCo' (*cf.*, Wedi 2014, Malardel *et al.* 2016), typically providing a resolution higher than the corresponding linear grid at the same spectral truncation. The model is stepped forward in time using a semi-implicit

time discretization for the faster (wave) processes. The model includes a realistic topography, state-of-the-art descriptions of the diabatic forcing processes, including shallow and deep convection, turbulent diffusion, radiation and five categories for water substance (vapour, liquid, rain, ice, snow). Full model documentation is available from: www.ecmwf.int/en/publications/ifs-documentation/.

## 2.2 General Description of the Study Period: Association of the vorticity variability with the MJO

As stated in the introduction, the vorticity is a key variable to be examined in this study. The vorticity field turns out to be rather "noisy", being dominated by smaller scales over the tropical region with the forecast correlation typically lost more than 60 % over a single day. For this reason, we judge that the vorticity field is rather an unreliable variable to diagnose over the tropics. The stream–function field is more robust, being obtained by applying an inverse-Laplacian to the vorticity, and by the nature of this inverse operator, this field is much smoother. This vortex structure is also expected to penetrate through the whole troposphere according to the free Rossby wave theory (*cf.*, Yano and Tribbia 2017). However in data analysis, the lower troposphere tends to be too noisy for identifying the MJO signature in the rotational wind field (vorticity) without a proper filtering or composite procedure (*cf.*, Wang *et al.* 2019). We focus on the tropopause level (150 hPa) in the following, because as it turns out, at this level, a coherent rotational flow field associated with the MJO is much easier to identify than the lower levels.

To see a clear association of the rotational wind field with the convective variability of the MJO, we show in Fig. 1 the time-longitude section averaged over 15S-15N for the outgoing longwave radiation (OLR) and the 150 hPa stream function (with the sign flipped for the southern hemisphere so that the anticyclonic vorticities are always treated as positive) for the four-month winter period (November 2016 – February 2017) from the ECMWF global analysis ("analysis" in short in the following), which is systematically adopted as an observational reference in the following. Here, data is plotted daily with the horizontal resolution of 2.5°. However, no filter is applied either in time or space. In the OLR field (Fig. 1(a)), three MJO events are identified over the Indian Ocean to the Western Pacific (90–180E) during this period, as identified as negative signals (in blue) stretching from the upper–left to the lower–right: the two major ones in December and in January–February. Another weak MJO event is identified during December–January.

In association with these three MJO events, high anticyclonic activities (positive signals, in red) over the Indian Ocean to the Western Pacific are identified (Fig. 1(b)), also propagating eastwards with a similar phase speed: the MJO constitutes of anticyclonic vortex pair in the upper troposphere propagating eastwards, as expected from the nonlinear free–Rossby wave theory: see Wang *et al.* (2019) for further discussions. Thus, according to this theory, these features need to be simulated in association with the MJO.

## 2.3 Choice of the forecast cases

Two forecast cases are mainly considered. Both cover one of the two most prominent MJO events during the northern winter 2016–2017 as seen in Fig. 1. The MJO event considered here corresponds to a low–skill event (F. Vitart, personal communi-

cation, March 2018) under dichotomic categorisation of the MJO forecast difficulties introduced by Kim *et al.* (2016), which are more difficult than the average. Here, a low–skill event is chosen for our experiments for an obvious reason that it is more challenging to forecast. As going to be seen below, operational control forecasts perform rather poorly, thus a question to be posed is: how can we improve them? Sensitivity experiments are chosen, as discussed in the introduction, with a hypothesis of the MJO as a nonlinear free–Rossby wave in mind. If this hypothesis is correct, we should obtain better forecasts by turning off selected physics.

The first forecast case (called "standard" in the following: Figs. 2(a) and 3(a)) is initiated on 19 January 2017 and run for 20 days. At this initial condition, convection associated with MJO is already fairly well developed over the Indian Ocean (Fig. 2(a)), and the key question is whether the model can maintain this convective system and also propagate eastwards as observed. On the other hand, from a dynamical point of view, this is before the anticyclonic activity begins to develop over the Indian Ocean (Fig. 3(a)). Thus the key forecast question is whether the model can predict the onset of this activity.

The second cases (called "extended" in the following: Figs. 4(a) and 5(a)) is initiated ten days earlier (9 January) than the standard case, and run for 40 days, except for the Mbb case (*cf.*, Table 2) runs for only 30 days. The initial condition corresponds towards the end of a previous MJO, and no mark of convective activity associated with the new MJO is yet to be seen over the Indian Ocean (Fig. 4(a)). Thus, a key operational challenge is to forecast the onset of convective variability associated with the MJO over the Indian Ocean. From a dynamical point of view, the vortex pair associated with the previous MJO is still well identified over the Western Pacific (Fig. 5(a)). Thus, another operational challenge is to forecast the continuous maintenance of this vortex pair, in association with a subsequent onset of another vortex pair over the Indian Ocean.

Finally, a single quasi-free forecast initiated on the 1 February 2017 is considered (QF). This is a moment that the vortex pair is fully developed over the given MJO event (Fig. 5(a)), although convection actually has already begun to fade out (Fig. 4(a)). Thus, this experiment examines whether it is possible to forecast the eastward propagation of this vortex pair even without convection. Table 2 describes the list of sensitivity experiments. As described in the Introduction, selective physics are turned off but only over the tropics, in the following experiments, by applying a factor, $1 - \cos^6 \phi$, on a physical term in concern with $\phi$ the latitude.

## 2.4 Analysis Procedure

### 2.4.1 OLR

We take the outgoing-longwave radiation (OLR) as a representative of the convective variability by following a standard approach in the literature. Here, however, special considerations are required with this variable, because within IFS, the longwave radiation (tagged as the "top net thermal radiation" $J/m^2$) is recorded as accumulated values. As a standard procedure at ECMWF, the emission rate is estimated from the accumulated values as a tendency over 24 hours. Since the outgoing longwave radiation is not one of the initialization fields, it is not included as an analysis field, either. As a result, "observational" OLR is, instead, estimated from the first 24-hour tendency of the operational daily forecasts. For this reason, even the initial 24-hour pattern correlation is noticeably less than the unity in the following presentations (Fig. 6(a) below). The OLR anomaly

**Table 2.** List of sensitivity experiments at TCo639 with 137 vertical levels: categories according to Table 1, the label used in the text, experiment description, and forecast cases (standard, extended).

| Category | Label | Experiment description | Forecast Cases |
|---|---|---|---|
| 1 | CF | Control operational forecasts | standard [19 Jan.–8 Feb.], extended [9 Jan.–18 Feb.] |
| 2 | Ma | OFF all the momentum dissipation (drag) tendencies in vertical eddy diffusion (including those in the boundary layer) and convection parametrization (shallow and deep) | standard [19 Jan.–8 Feb.] |
| 2 | Mbe | OFF momentum dissipation tendencies due to vertical eddy diffusion only | standard [19 Jan.–8 Feb.] |
| 2 | Mbb | OFF momentum dissipation tendencies due to vertical eddy diffusion (boundary layer below 800hPa) | standard [19 Jan.–8 Feb.], extended [9 Jan.–8 Feb.] |
| 2 | Mbc | OFF momentum dissipation tendencies due to convection parametrisation (shallow and deep) | standard [19 Jan.–8 Feb.] |
| 2 | Mbd | OFF momentum dissipation tendencies due to convection parametrisation (deep only) | standard [19 Jan.–8 Feb.] |
| 2 | Mbs | OFF momentum dissipation tendencies due to convection parametrisation (shallow only) | standard [19 Jan.–8 Feb.], extended [9 Jan.–18 Feb.] |
| 2 | Mbde | OFF momentum dissipation tendencies due to vertical eddy diffusion and convection parametrisation (deep only) | standard [19 Jan.–8 Feb.] |
| 2 | Mbse | OFF momentum dissipation tendencies due to vertical eddy diffusion and convection parametrisation (shallow only) | standard [19 Jan.–8 Feb.] |
| 3 | NQ | OFF physical tendency for the temperature (entropy) (due to shallow and deep convection, radiation and cloud phase changes) | standard [19 Jan.–8 Feb.] |
| 4 | QF | OFF all physical tendencies as above for both momentum and temperature | standard [19 Jan.–8 Feb.], extended [9 Jan.–18 Feb.], 20 days [1–21 Feb.] |

is defined as a deviation from the climatology. Here, the climatology is defined as an average over the years 1979–2009 for each given calendar day.

### 2.4.2   150–hPa Stream Function

For examining an association of MJO with the vorticity field, *or* rotational flow, we take the 150–hPa stream function, as already discussed in the beginning of Sec. 2.2.

### 2.4.3   Verification

In the following, the forecast performance is evaluated by inspecting time–longitude section of OLR and the stream function averaged over 15S–15N, considering the fact that the MJO is a longitudinally–propagating feature. When latitudinal interactions between MJO and higher–latitude Rossby waves are in concern, time–latitudinal sections are examined instead. In the present study, we emphasise an importance of the visual inspection of the forecast performance to compare it with the analysis. In the following, very specific descriptions of the forecast behaviours in comparison with the analysis or a control forecast will be presented, because we believe that these details are keys to understand the actual processes simulated by these forecasts.

As a basic point of reference, the correlation is computed between the analysis and a forecast over the longitudinal range of 0-180E between 15S and 15N. This correlation will be referred as a *pattern correlation* in the following. We adopt this measure, because it is a straight manner of comparing the two fields (analysis and forecast) over the tropics without imposing our prejudices of expectations.

Additionally, evolutions of forecasts in the phase space of the real–time multivariate MJO (RMM) index pair (Wheeler and Hendon 2004) are also presented for selective cases. Here, the RMM index pair is evaluated by projecting the temporal anomaly defined as a deviation from an average over a forecast period. Note that unlike the pattern–correlation analysis, the RMM measures a forecast skill in respect to a prescribed field pattern. This design exactly becomes a key limitation of RMM (*cf.*, Straub 2013).

## 3   Results

### 3.1   Summary of forecast experiments: the pattern–correlation analyses

The time series of pattern correlations between the forecasts and the analysis in Fig. 6 summarise the experiment results. The anomaly field is considered for the statistics of the OLR, whereas the zonal mean is taken out from the 150–hPa stream function. A first step of verifying the performance of the sensitivity experiments would be to examine how well the convective variability associated with the MJO is predicted by these experiments. The pattern correlations between the simulated OLR and the analysis are shown in Fig. 6(a). The same is shown in Fig. 6(b) for the rotational–wind field (150–hPa stream function). Fig. 6(c) is the same as Fig. 6(b), but focuses on the role of convective frictions (*cf.*, Sec. 3.3.2 below).

As another summary for the forecast performances, Fig. 7 present RMM analyses for some selective cases. Here, (a) and (b), respectively, show the evolution trajectory of the analysis data on the RMM phase space over the standard and extended forecast periods. Evolution of the MJO is represented by a counter–clockwise movement of a trajectory in this phase space, with an initial point marked by a red circle, as seen in both frames. Note that although the extended forecast period contains the standard forecast period as a part, the two trajectories for the ERA5 analysis do not match exactly over the same period due to the different definitions of the temporal anomaly used (defined relative to an average over a selected forecast period). These two trajectory patterns are to be compared with those of sensitivity experiments and control forecasts as a verification. However, afore–mentioned mismatching fundamentally limits the applicability of the RMM analysis in the following.

The remainder of this section proceeds as follows: morphological behaviours of the control forecasts are carefully described in the next subsection (Sec. 3.2), because they provide baselines for interpreting subsequent runs turning–off selected physics. The following two subsections (Secs. 3.3 and 3.4) look for improvements of MJO forecasts by removing momentum dissipation as well as diabatic heating effects, as would be expected from the free nonlinear Rossby–wave theory. As it turns out the performance of the MJO forecasts does not depend on these choices of physics in any consistent manner: less momentum friction does not necessarily lead to a further improved MJO forecast, but the skill and MJO propagation sensitively depends on the type of dissipation turned off. Effects are hardly additive, either, but clearly nonlinear interactions are going on between the physics. Thus, against the original motivation stated in the introduction, the main purpose of these two subsections becomes a report of these forecast sensitivities in more detail. Careful descriptions will also reveal that improvements of the MJO forecast is hardly a monotonic measure: certain aspects are improved, but often associated with deterioration of other aspects. Sec. 3.5 focuses on the model performance on simulating interactions between the MJO and higher–latitude Rossby–wave activities. Here, we find a consistent tendency that the model simulates those interactions features identified in the analysis rather well, although some sensitivities inevitably emerge.

## 3.2 Control Forecasts (CFs)

### 3.2.1 Standard 20–Day Control Forecast

With the standard 20-day control forecast (CF), the initial 0.7 pattern correlation of OLR with the analysis linearly decreases to 0.5 approximately at the end of the forecast (thin black curve in Fig. 6(a)). Inspection of the time-longitude section (Fig. 2(b)) reveals that although the convective variability is persistent in the simulation, it is too stationary (lack of propagation), and as a result it loses a pattern correlation with the analysis with time (*cf.*, Fig. 2(a)).

The standard CF presents a rather high pattern correlation of the 150–hPa stream-function with the analysis above 0.8 for the first 16 days (thin black curve in Fig. 6(b), (c)). However, this high pattern correlation turns out to be rather misleading, because a direct inspection of the time-longitude plot (Fig. 3(b)) reveals that the predicted stream–function signal is much weaker than analysis (Fig. 3(a)). Onset of the anticyclonic vorticity signal centered around 100E on 29 January is correctly predicted, leading to a high pattern correlation, but with a much weaker amplitude, and the signal suddenly dies out on 4 February associated with a sudden drop of the pattern correlation.

As expected from the description so far, the MJO signal as defined by RMM index (Fig. 7(c)) rapidly decays in the standard CF, and a forecast skill is totally lost in less than 10 days.

### 3.2.2  40–Day Extended Control Forecast

When the experiments are initialized 10 days earlier (9 January), the forecast is expected to be harder, because it corresponds to a final stage of the previous MJO, and a next MJO to be predicted is not yet initiated (*cf.*, Fig. 4(a)). The pattern correlation of OLR gradually decreases to 0.4 over 20 days with CF (thick black curve in Fig. 6(a)). However, from this point, the pattern–correlation value begins to gradually recover, and it exceeds that of the standard 20-day forecast on 2 February, and increases to above 0.6 by 4 February.

Some possible interpretations are inferred from the time-longitude section (Fig. 4(b)). The last phase of the previous MJO consists of a westward propagating cloud cluster over the Western Pacific, partially driven by the linear Rossby wave dynamics. In the extended CF, this westward propagating cloud cluster continues to propagate into the Indian Ocean although it dissipates out in analysis. The continuous westward propagation effectively simulates the initiation of the new MJO, as observed. The termination of this cloud cluster on 26 January coincides with an initiation of a new cloud cluster to its east side. The new cloud cluster is also more persistent than the observed counterpart, that in turn, contributes to a recovery of the pattern correlation. It is speculated that the persistence of this cloud cluster is helped by a persistent anticyclonic signal over the same region, successfully predicted albeit with a 4-day delay of onset (Fig. 5(b)). The simulation predicts an initiation of another convectively active phase on 11 February, as observed. However, this convective variability turns out to be more active and persistent than observed.

According to Fig. 7(d), the MJO signal defined by the RMM initially decays rapidly over the first 5 days. However, the forecast skill gradually recovers towards the end of the forecast by following a circle marked in the phase space (corresponding to a standard deviation of climatological RMM index pair).

## 3.3  Forecasts Sensitivities on Friction

Forecast performance sensitively changes by turning off some physical processes. We focus mostly on the standard 20–day forecasts first to elucidate various aspects, then briefly remark on the 40–day extended forecasts.

### 3.3.1  Momentum Dissipation

Performance of the forecasts for the MJO rotational field sensitively depends on the choice of momentum dissipation terms. This subsection discusses the overall aspect. The next subsection focuses more specifically on convective friction. A first case to be considered is when the total tendency for the momentum dissipation (both eddy diffusive and convective: Ma) is turned off. The time-longitude section (Fig. 2(c)) shows that the eastward propagation structure of convection is better simulated than by CF. However, convection also becomes too strong compared to the analysis. More significantly, a clear-sky area (60-70E) behind the MJO convective variability seen in the last 8 days in the analysis, but absent in CF, is successfully predicted in this

case. The RMM analysis (Fig. 7(e)) also shows that the Ma run evolves around a well–defined counter–clockwise circle with a large radius in the phase space.

Turning off the vertical–eddy momentum dissipation both totally (Mbe: Fig. 2(d)) and only in the boundary layer (BL, below 800 hPa: Mbb: thin blue curves in Fig. 6(a) and (b); Fig. 2(e)) leads to similar results. Inspection of their time-longitude plots show that the eastward propagation tendency is better simulated by these two cases (Mbe, Mbb) than when the momentum dissipation (drag) is totally turned off (Ma: Fig. 2(c)). Intensity of convection also reduces to a reasonable level, also presumably contributing to slow down the propagation (*cf.*, Seo *et al.* 2009).

Inspection of the time-longitude sections of the 150–hPa stream function for those cases reveal that the anticyclonic variability associated with the MJO event is better simulated by these cases than CF: the emission of the Rossby wave energy from west during 22–28 January is speculated as a major source *e.g.*, for initiating the anticyclonic signal associated with the MJO by the time-longitude plots (Fig. 3(c) for Ma). However, the wave structure to the west of the MJO anticyclone is exaggerated compared to the analysis: it may be interpreted as a westward propagation of a free Rossby wave. A similar feature in the rotational–wind field as in Ma is also identified with the Mbb (Fig. 3(d)), but in a more intermittent manner. The forecast performance of these cases for the 150–hPa stream function in terms of the pattern correlation is, however, not any better than the CF case as seen in Fig. 6(b).

### 3.3.2   Convective Friction

Turning off the convective friction tends to prolong the predictability of the MJO signal substantially as seen with the rotational wind field in Fig. 6(c) for the standard 20-day forecasts: a pattern correlation is typically maintained at a relatively high value (*ca*., 0.8) until the end of the forecast, in contrast to a sudden drop of the pattern correlation with the CF (down to *ca*., 0.4) over the last 4 days.

When the convective friction is totally turned off (Mbc: pink in Fig. 6(c)), the pattern correlation is occasionally higher than the CF case even during the first 16 days of the forecast. Inspection of the time-longitude section (Fig. 3(e)) shows that the predicted MJO signal in rotational–wind field is also comparable to the analysis (Fig. 3(a)). When only the shallow convective friction is turned off (Mbs: red in Figs. 6(b) and (c)), the pattern correlation remains higher during the last phase of the forecast than when the convective friction is totally turned off. Time-longitude section (Fig. 3(f)) reveals that in this case, the anticyclone signal over 100-120E persists throughout the experiment without a break over the period of 21-27 January as observed.

In contrast, when only the deep convective friction is turned off (Mbd: blue in Fig. 6(c)), the forecast performance substantially deteriorates in the last phase. The deterioration is associated with an over-enhancement of the anticyclonic signal over the last phase (29 January to 8 February: Fig. 3(g)). When both deep-convective and boundary-layer frictions are turned off (Fig. 3(h): Mbde), the second anticyclonic variability event is too strong, and too spread to the west. When shallow-convective and boundary-layer frictions are turned off (Fig. 3(i): Mbse), anticyclonic variabilities dramatically weaken. Especially, the second anticyclonic variability is too weak and too short: terminated 4 days before the end of the forecast.

Thus, less momentum friction does not positively contribute to the MJO forecast in any consistent manner. These modifications, rather, suggest that effects of turning off the momentum dissipation are not additive, suggesting that some nonlinear interactions are going on.

### 3.3.3    40–Day Extended Forecasts

With the extended forecast when the shallow convective friction is turned off (thick red curve in Figs. 6(b) and (c), Figs. 4(c) and 5(c): Mbs), the behaviour of the 150–hPa stream function (Fig. 5(c)) is overall similar to that of the standard CF, except for some precursors for the anticyclonic signal leading to the new MJO event and a re-development of the anticyclonic variability towards the end of the forecast. When the boundary-layer friction is further turned off (30 days in blue, Fig. 6(b), Figs. 4(d) and 5(d): Mbb), the initial anticyclonic variability continues about 6 days longer than observed, and the second anticyclonic variability is also initiated 1-2 days later than observed (Fig. 5(d)). Its precursor, albeit weak, already has a good pattern correlation with the analysis.

These extended forecasts may be overall interpreted to suggest that turning off the momentum friction contributes to an improvement of the MJO forecast in general. However, a further removal of the momentum friction in the boundary layer (Mbde and Mbse, light blue and orange in Fig. 6(c), respectively) slightly reduces the forecast performance.

An initial phase of forecast of the rotational wind field (vorticity field) is easier when the experiment is initiated 10 days earlier than otherwise, because the initial condition corresponds to the maximum of the anticyclone signal (centered at 100-120E) associated with the previous MJO (Fig. 5(a)). A gradual decay of the pattern correlation (with this anticyclonic signal) over the next 4 days is reasonably predicted by CF (Fig. 5(b)), as well as the cases without shallow convective friction (Fig. 5(c): Mbs) as well as without boundary-layer momentum dissipation (Fig. 5(d): Mbb).

Further analysis suggests that the 40-day extended CF simulates the rotational field associated with a MJO rather for a wrong reason: a dipolar vortex structure, constituting an analogue to analytical nonlinear *modon* solution is formed by the northern-hemisphere anticyclone with a well–isolated cyclone further north rather than with a southern-hemisphere counterpart. The same interpretation also applies to the Mbs case.

### 3.4    Free–Dynamics Experiments

This subsection gradually turns off more forcing and dissipation terms so that the system may gradually approach to a state of free dynamics.

We first turn off diabatic heating totally (NQ) so that the vortex dynamics is no longer coupled with convection. Without surprise, the pattern correlation steadily decreases with time approximately linearly to 0.2 towards the end of the standard forecast. The inspection of the time-longitude section of 150–hPa stream function (Fig. 3(j)) shows that the rotational wind field at this level decays fairly rapidly without diabatic heating, but leaving a small-amplitude wave field. It may be worthwhile to emphasise that the decay process of the anticyclonic signal from the previous MJO is fairly realistic in this forecast, though arguably slightly too fast. A subsequently-generated weak wave field may also be worthwhile to discuss: the cyclonic signal

centered around 220-250E amplifies realistically as observed, then it leads to a westward propagation, presumably as a free linear Rossby waves, which turns into a anticyclonic signal around 170E and continues to propagate westward. On 31 January, the anticyclonic signal arrives 100E. We speculate that it contributes to a significant recovery of the pattern correlation (*ca.*, 0.6 from *ca.*, 0.2 two days earlier). Those relatively positive evaluations of the NQ forecast is supported by the RMM analysis (Fig. 7(f)): it evolves around a well–defined counter–clockwise circle, albeit with a relatively small radius.

When the momentum friction is further turned off (QF), the OLR signal decays over the first few days (about four days: Fig. 4(e)) with the 40–day extended forecast. Though some pattern correlations persist beyond this point, that is achieved only by a very weak OLR signal predicted. With the standard forecast of QF (thin green curves in Fig. 6(a) and (b)), rather unintuitively (despite the lack of momentum dissipation), the westward propagating Rossby-wave signal decays much faster and the amplitude is weaker (Fig. 3(k)) than the case without turning off the momentum friction (NQ), say, by a factor of three. As a result, the pattern correlation with the analysis also becomes slightly smaller (by 0.1-0.2). A similar behaviouris also seen with an extended run (Fig. 5(e): QF).

A final experiment to test the idea of free MJO dynamics is initiated on 1 February 2017 (QF), when a vorticity pair associated with the MJO is already fully developed, as seen in analysis (Fig. 8(a)). Thus, this experiment examines whether it is possible to forecast the eastward propagation of this vortex pair even without convection. At this phase, convection is no longer very active. The quasi–free forecast of 150–hPa stream function for 20 days is shown in Fig. 8(b). The result is rather disappointing in the sense that the vortex pair rapidly dissipates over the first few days. It suggests that the model is still not dissipation-less enough as we intend. Nevertheless, a rather surprising behaviour is an eastward propagation of the vortex pair as expected for nonlinear solitary Rossby waves, and opposite to a sense of propagation direction expected for linear Rossby waves. However, the propagation speed of this decaying vortex pair is much faster than that is found in the analysis.

## 3.5 Intrusion of an Extratropical Rossby Wave Train? Standard 20–Day Forecasts

Some studies (Hsu *et al.* 1990, Gustafson and Weare 2004, Ray and Zhang 2010, Ray and Li 2013, Zhao *et al.* 2013, Wang *et al.* 2019) suggest that an intrusion of a Rossby-wave train from the northern hemisphere to the tropical region can initiate a MJO.

The analysis of standard 20 day forecast period finds such an example over 20-27 January, as depicted in a time-latitude section for the 150–hPa stream function averaged over 20E–60E (Fig. 9(a)): a negative stream–function signal (cyclone) arrives from 80N to 30N by taking about 5 days. An inspection of this time-latitude section gives an impression that the arrival of this signal to 30N helps to re-vitalise and sustain longer the anticyclonic signal centred at 15N. Since its eastward extension is considered the MJO, it leads to an interpretation that the arrival of such a Rossby-wave train helps to initiate the anticyclonic variability (vortex pair) associated with the MJO.

However, the forecast experiments tend to not favour the above interpretation in terms of the Rossby–wave train. To see this point, the performance of the CF for the same period is, first, shown in Fig. 9(b): the arrival of the Rossby-wave train appears to enhance the anticyclone over the same longitudinal range centred at 15N to a degree more than in analysis. However, as a

separate time-longitude section shows (Fig. 3(b)), the anti-cyclonic signal associated with MJO decreases faster than observed
over the same period with CF.

Three additional experiments (NQ, QF, Ma) provide further insights (Figs. 9(c), (d), (e)): The first is a case with all the diabatic heating (radiation, convection, cloud physics) turned off (Fig. 9(c): NQ). The second case is with both diabatic heating and all the momentum dissipation (vertical eddy transport and convection) turned off (Fig. 9(d): QF). In both cases, the arrival of the Rossby-wave train with a cyclonic signal to the subtropics (30N) is well simulated, and the resulting cyclone signal along
30N is more persistent than in CF, and even more so than in the analysis. Presumably, the absence of the momentum dissipation helps to amplify the cyclone signal with time along 30N (QF), although it is less persistent than the case without turning off any momentum friction (NQ). In both cases, a further induction of the anticyclone signal along 15N, though identifiable, much weaker than the CF case, and it totally disappears after 3 February. Finally, when all the momentum friction is turned off, but the diabatic heating is maintained (Fig. 9(e): Ma), the cyclonic signal intruding into the subtropical region ($ca.$, 30N) from the
higher latitudes becomes even weaker than in the analysis. The anticyclone anomaly is induced along 15N in a realistic manner without further amplification as with the CF case.

The predictions of the rotational field in standard 20–day forecasts are overall reasonable in patterns, but larger errors in amplitude. An impression is that the MJO dipole is less isolated than in the analysis, thus the internal (nonevanescent) wave structure leads to westward propagation (or stalled) rather than eastward.

**4   Discussions**

A main motivation for the present study has been to examine the extent that the MJO can be simulated with a relatively frictionless (physically unforced) setting, being consistent with the proposed free nonlinear Rossby–wave theory for the MJO by Wedi and Smolarkiewicz (2010), Yano and Tribbia (2017), Rostam and Zeitlin (2019), and Wang $et$ $al.$ (2019). The MJO forecast does indeed improve when the momentum dissipation is totally removed (Ma: $cf.$, Figs. 2(c) and 3(c)). However,
the tendency is hardly consistent: the degree of forecast improvements sensitively depends on the choice of momentum–dissipation terms turned off. The effects are hardly additive, either, and clearly certain nonlinear interactions are going on. Most disappointingly, when all the dissipation and forcing terms both for the momentum and the entropy are turned off (QF), the features associated with MJO disappear rather rapidly (Fig. 3(k)). Thus, the present study does not support the proposed free nonlinear Rossby–wave theory in any consistent manner. Details on the forecast behaviour based on the choice of physical
configurations of the model have been carefully documented to record the unexpected but nevertheless important impact on MJO forecast skill.

There are several lessons to learn from the present sensitivity exercise. The first is the importance to closely evaluate the details of sensitivities of physical processes for the MJO. In typical mechanism–denial studies ($e.g.$, Kim $et$ $al.$ 2011, Ma and Kuang 2016), a physical process in concern is either totally turned off or maintained. If we would have followed such an
approach, the improvement of MJO forecasts by totally removing momentum dissipation (case Ma) would have simply been interpreted as a positive result for supporting a free–wave theory. However, in the present study, the momentum dissipation

processes, arising from various different physical mechanisms, are turned off selectively to verify this initial finding in a more solid manner. As it turns out, the sensitivities of MJO on momentum dissipation processes are not that simple. Though we are short of making any definite conclusions from our sensitivity study of the MJO on the momentum dissipation processes, the study suggests a critical importance of examining the physical sensitivities of a phenomenon with more detail rather than simply switching off the entire physical mechanism as has been done in past sensitivity studies. Second, after examining the forecast results closely, we have realised that it is not quite straightforward to simulate a free wave dynamics expected from the theory with a complex state-of-the-art global model, as originally intended. We conclude that this difficulty stems from a need for maintaining a realistic background state at the same time (*cf.*, Ma and Kuang 2016).

Discussions on some specific runs make this point clearer: with the quasi-free 40-day extended forecast (QF: Fig. 5(e)), the pre-existing anticyclonic variability over 100-150E persists almost as long as observed (7 days), albeit with weak amplitude. A weakly eastward tendency, being consistent with the nonlinear free-wave theory, may also be noticed in this simulation. In the standard 20-day forecast case, only a reminiscence of the anticyclone signature from the previous MJO event is found around 120E initially in analysis, and this feature disappears in less than two days (Fig. 3(b)). Note that no convective variability is found at the vicinity to this longitude at the initial time of this forecast period (Fig. 2(a)). The quasi-free forecast (QF) maintains anticyclonic variability longer than in the analysis albeit with a weaker amplitude (Fig. 3(k)).

We interpret these rather subtle results with the quasi–free (QF) forecast experiments as a demonstration of the difficulties for realising a "realistic" free–dynamics experiment. The main problem with the QF forecasts in the present study is a fact that by practically turning off "all" the physical forcings, the basic state of the model also breaks down very rapidly, thus a proper background state that may support a free–dynamics MJO is also lost very rapidly. It also follows that a free MJO mode also dissipates out very rapidly. A more appropriate manner of performing free–dynamics experiments would be to maintain a background state with full physics in place, but to introduce quasi-free dynamics only to a perturbation component. The basic idea of this strategy may be understood in analogy with standard perturbation analyses. However, in the present case, perturbations must be treated in a fully nonlinear manner, to be consistent with our anticipation that the MJO is a fully nonlinear construct. A brute–force approach of nudging the model towards a climatology (*e.g.*, Ma and Kuang 2016) may be valid, but only when the given climatology is a correct "background state" to maintain. A more delicate procedure is required, for example, by using the emerging modelling infrastructure described in Kühnlein *et al.* (2019), so that any constraints on the evolving nonlinearities are removed.

The present study further suggests that the MJO predictability sensitively depends on the choice of the initial condition in a rather unintuitive manner, but being consistent with a clear distinction between high– and low–skill MJO events identified by Kim *et al.* (2016): longer forecasts from an earlier phase of the MJO may not be harder than a shorter one from a later phase. In the present study, the standard forecasts are initiated (on 19 January) from an early stage of an MJO already present, thus a successful forecast would simply capture the subsequent development and propagation of this MJO. On the other hand, the 40–day extended forecasts are initiated 10 days earlier towards the end of a previous MJO event. Presumably, the latter is harder to forecast the MJO evolution, especially an onset of a new MJO. However, an inspection of the time-longitude section suggests a different picture: the longer 40-day extended forecasts tend to regenerate the MJO signal towards the end of the forecasts,

and the recover the forecast capacity. In some cases, their performance becomes even better than the shorter standard 20–day forecasts initiated 10 days later in terms of the pattern correlations of the OLR and the 150–hPa stream function (Fig. 6).

As Nakazawa (1988) originally pointed out, the MJO typically constitutes a modulation of the westward-propagating cloud clusters of few-hundred km-scale. The 9 January, the initiation time of the 40-day extended forecasts, corresponds towards the end of the previous MJO event, and also a moment that the last cloud-cluster over the Western Pacific begins to propagate westwards, that marks the end of this MJO event (Fig. 4(a)). In the 40–day extended CF (Fig. 4(b)), this westward-propagating cloud cluster does not die out as observed, but continues to propagate westwards to the Indian Ocean, which marks an initiation of a new MJO under this forecast. Though the predicted new MJO weakens out at a middle, we note a recovery of the signal

towards the end of the event. These initial condition sensitivities of the MJO forecasts point to a simple fact that an onset as well as evolution of a MJO should not be considered as an isolated event, but better interpreted as a part of chain of processes in the atmosphere. It also points to an importance of better understanding detailed processes associated with the MJO, in the present case, those of the westward–propagating cloud clusters. Standard MJO indices (*e.g.*, RMM) fail to depict those critical details (*cf.*, Straub 2013).

The present study has also elucidated active interactions of MJOs with higher–latitude Rossby–wave activities (Fig. 9). Inspections of the latitude–time sections suggest that the performance of the MJO forecasts appears, at least partially, to be helped by the successfully simulated interactions of the MJO with the higher-latitude Rossby waves (Rossby–wave trains).

The MJO forecast problem is often reduced to that of convection parameterizations (*e.g.*, Hirons *et al.* 2013a, b, Jiang *et al.* 2015, 2020, Pilon *et al.* 2015). However, improvement of the MJO forecast, along with the many other forecast issues, is

460 not a matter of fixing a single physical scheme. Rather we need to examine a forecast model as a whole with its interacting physics for achieving this goal. The present model–sensitivity study has exposed the importance of physical processes other than convection on maintaining a realistic tropical mean state and on MJO forecast skill.

The complex behaviour of the IFS model sensitively depending on the choice of physics turned off, as identified in the present study, should be emphasised in its own right. For example, the role of momentum friction, in general, is not simply

favourable or unfavourable for MJO forecasts. The behaviour sensitively depends on the precise type of momentum friction being turned off. In other words, the operational model behaviour is not decided by a dichotomy of with or without friction, as typically assumed in theoretical as well as in some model sensitivity studies. By reporting the details of these physical sensitivities on the MJO forecast, the present study strongly suggests a need for theoretical investigations that are much more closely tied to the actual operational formulations of physical parametrisation and their impact on the mean circulation rather

than merely modulating the (MJO) anomaly.

*Author contributions.* Forecast experiments were performed by NPW, and the graphic analyses were mostly performed by JIY. The manuscript was developed by closely analysing and discussing the results by the two authors

*Competing interests.* There is no competing interest with the present work.

*Acknowledgements.* Discussions with Peter Bechtold and Fredric Vitart are much appreciated. Fredric Vitart has also kindly provided us with a RMM plotting package. A bulk part of the work was performed when the first author was visiting ECMWF during February-March 2018. Special thanks are due to Peter Haynes, who made unusual effort for assisting us in revision process as the editor in charge. He also suggested us several key references.

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

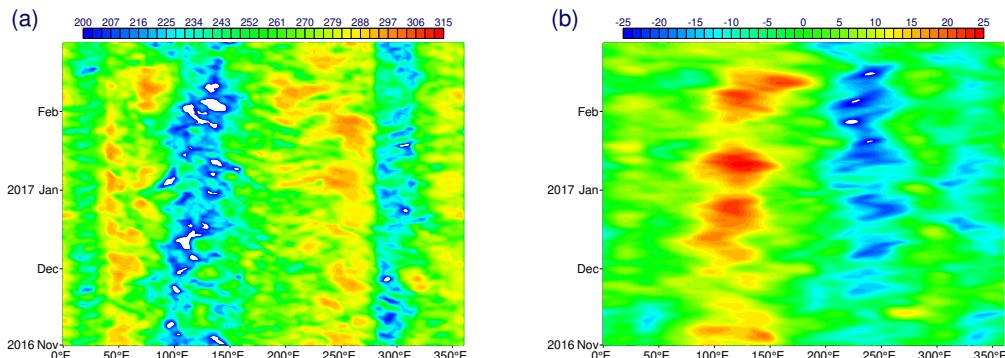

**Figure 1.** Time–Longitude sections of the ECMWF analysis averaged over 15S–15N for (a) OLR [K, as equivalent black–body temperature] and (b) the stream function [1/s] at 150hPa for the four–month winter period of 2016–2017. In averaging the stream function, the sign is flipped for the southern hemisphere.

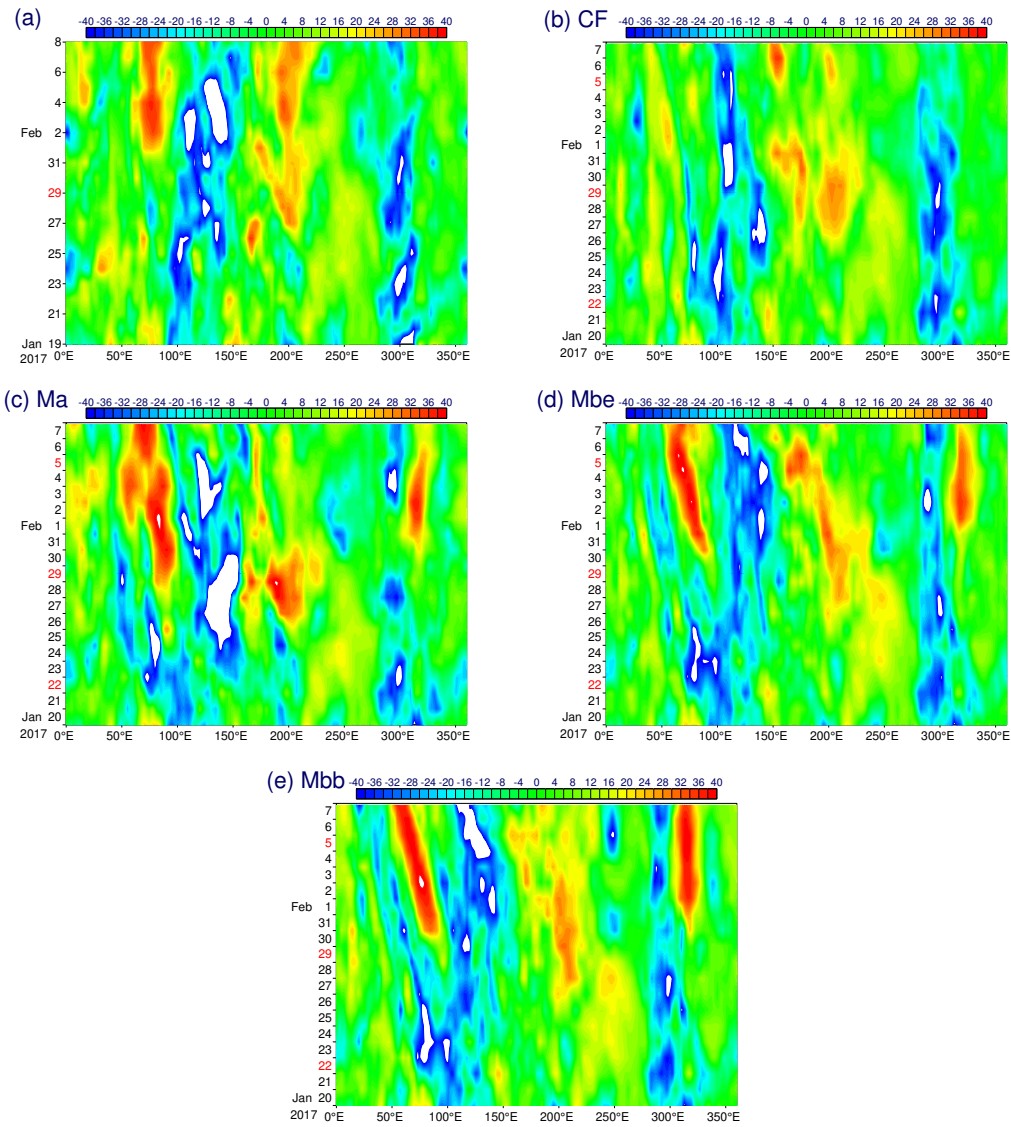

**Figure 2.** Time–Longitude sections averaged over 15S–15N of OLR for the standard 20–day forecast case: (a) analysis, (b) CF, (c) Ma, (d) Mbe, and (e) Mbb.

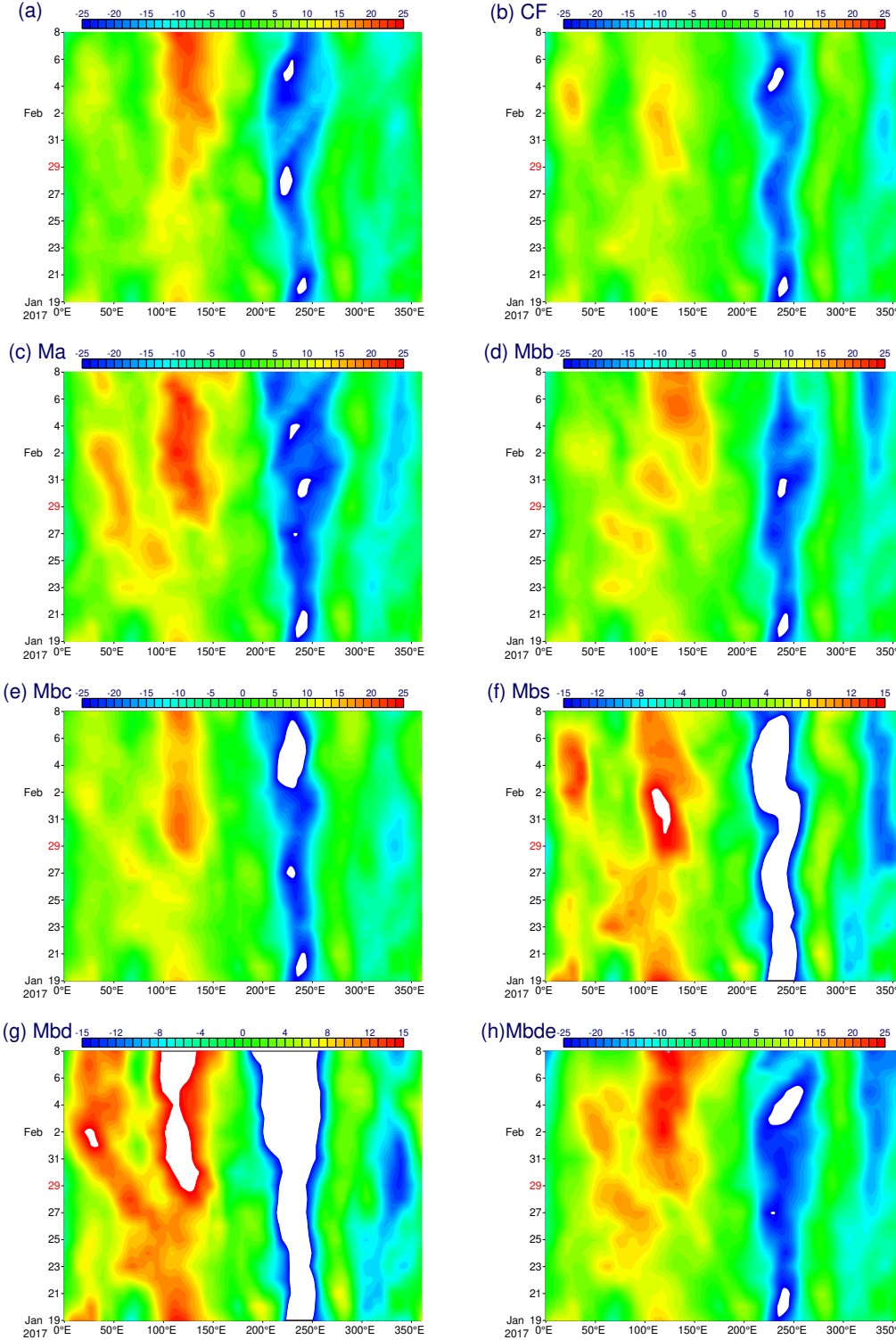

**Figure 3.** [see the Caption in next page]

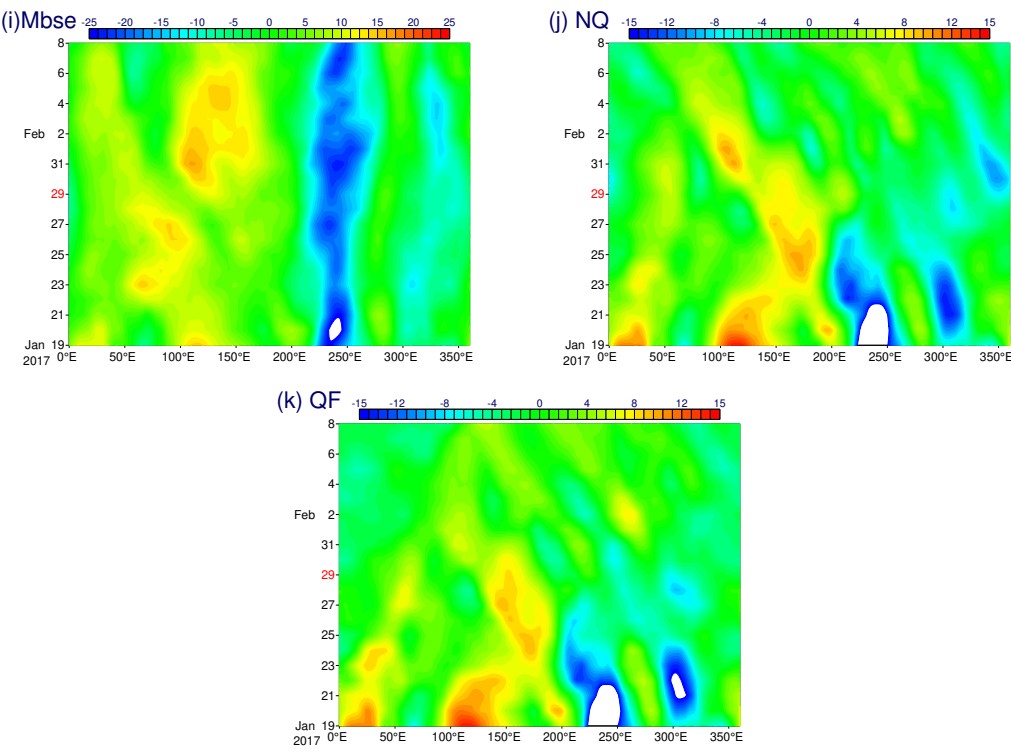

**Figure 3**: Time–Longitude sections averaged over 15S–15N of the 150hPa-level stream function for the standard 20–day forecast case: (a) analysis, (b) CF, (c) Ma, (d) Mbb, (e) Mbc, (f) Mbs, (g) Mbd, (h) Mbde, (i) Mbse, (j) NQ, (k) QF.

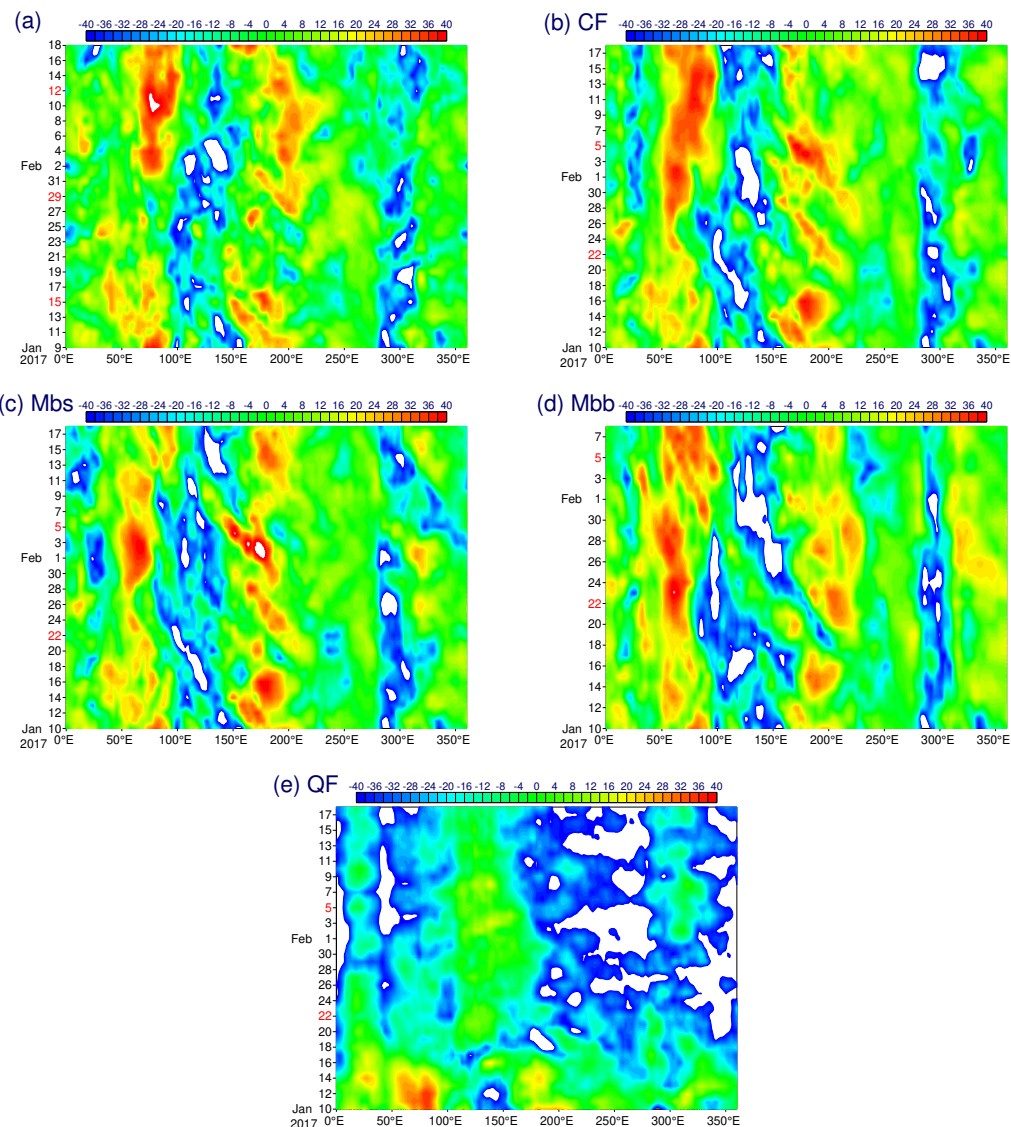

**Figure 4.** Time–Longitude sections averaged over 15S–15N of OLR for the 40–day extended forecast case: (a) analysis, (b) CF, (c) Mbs, (d) Mbb, and (e) QF. Note that the Mbb case (d) exceptionally runs only for 30 days.

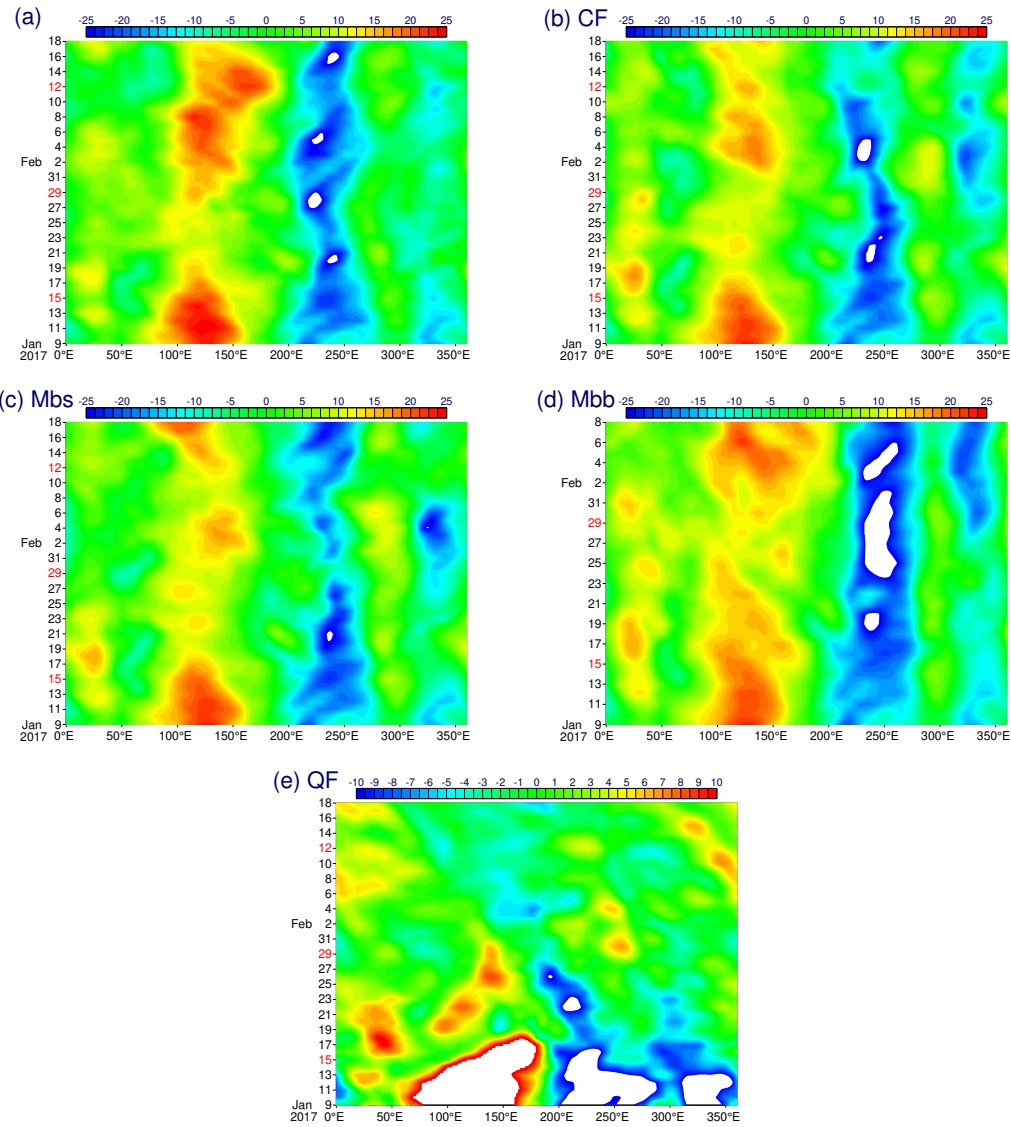

**Figure 5.** Time–Longitude sections averaged over 15S–15N of the 150hPa-level stream function for the 40–day extended forecast case: (a) analysis, (b) CF, (c) Mbs, (d) Mbb, and (e) QF. Note that the Mbb case (d) exceptionally runs only for 30 days.

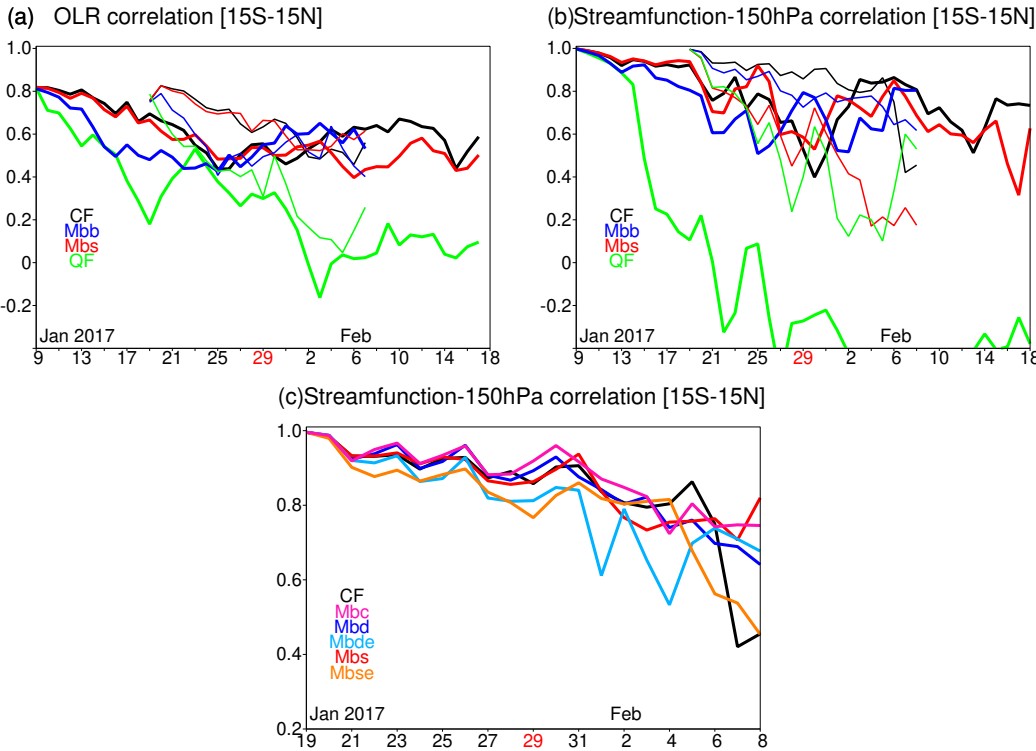

**Figure 6.** Time series of pattern correlations between the forecasts and the analysis over the longitudinal bands between 15S and 15N for (a) OLR and (b, c) the 150hPa-level stream function. The cases shown in (a) and (b) are: CF (black), Mbb (blue), Mbs (red), and QF(green); in (c): CF(black), Mbc (pink), Mbs (red), Mbd (blue), Mbde (light blue), and Mbse (orange). The standard 20–day and the 40–day extended forecasts are, respectively, shown by thin and thick curves.

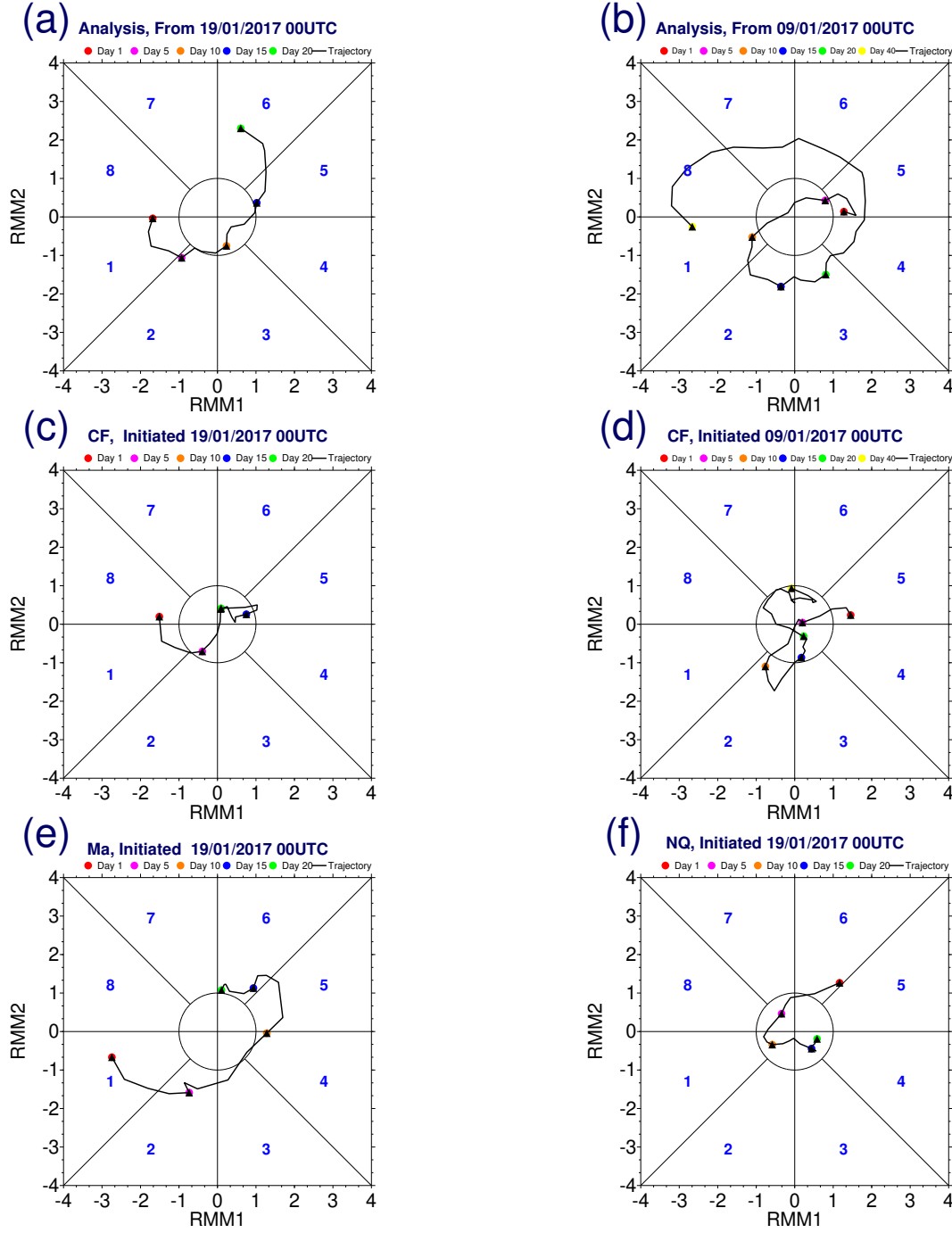

**Figure 7.** RMM plots for the analysis (a, b), for control forecasts (c, d) for the standard (a, c) and the extended (b, d) forecast cases. Also Ma (e) and NQ (f) cases for the standard forecast.

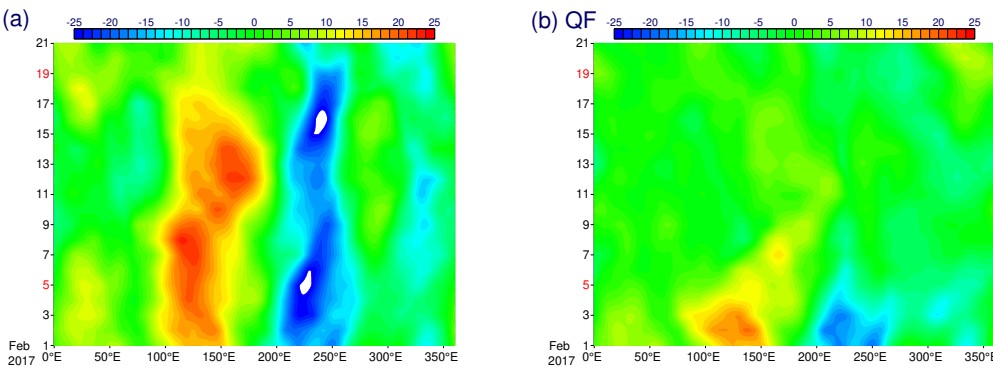

**Figure 8.** Time–longitude sections of 150–hPa stream function along the equator (15S–15N) for the 20–day period from 1 February: (a) analysis and (b) QF.

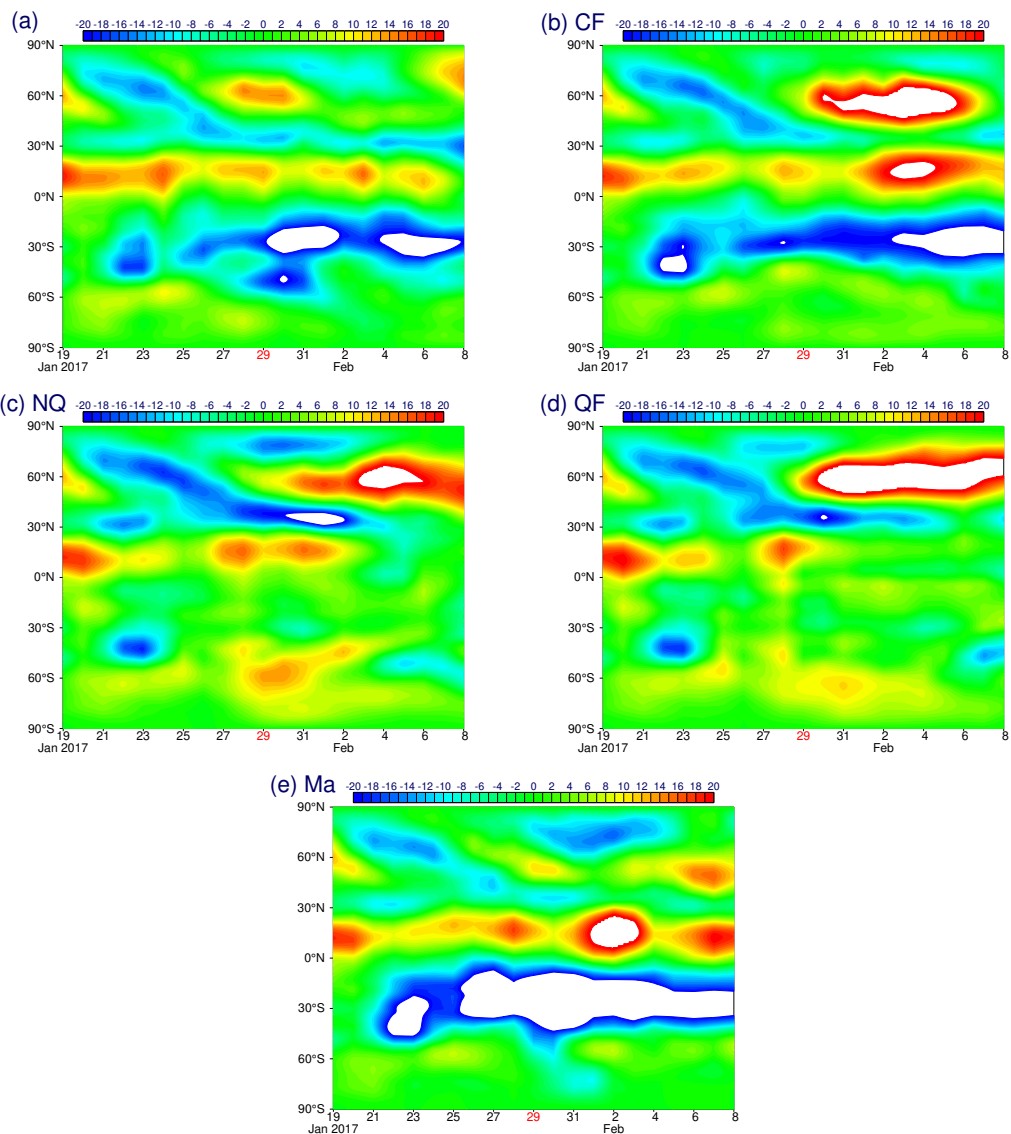

**Figure 9.** Time–latitude sections of 150–hPa stream function averaged over 20–60E: (a) the analysis for the standard 20–day forecast period, and standard 20–day forecasts with (b) CF, (c) NQ, (d) QF, and (e) Ma.