# Peer review of "Sensitivities of the MJO Forecasts on Configurations of Physics in the ECMWF Global Model"

_Atmospheric Chemistry and Physics, 2020_

## Referee Comment (RC1) · Anonymous Referee #1 · 3 Mar 2020

This study tries to answer the scientific questions related to 1) the possibility of interpreting the MJO as a free nonlinear Rossby wave through the sensitivity test to the diffusion terms, and 2) the MJO initiation process. Although a lot of sensitivity tests are done in this study, I think the overall interpretation and description are insufficient. Also, it was hard to figure out the scientific questions and related conclusions. I mention below in detail and hope to get more clear information in the revised manuscript.

Major comments

1. Please explain the physical meaning of the difference between the results in Section 3. Figures in Section 3 show some differences between the experiments. For exam-

ple, what is the role of momentum dissipation and vertical eddy diffusion to make the difference (Figs. 4, 5)? How do they affect the simulation of MJO propagation and magnitude? Also, in my opinion, the convection in Fig. 4a is stronger than those in Figs 4b, c. Because more momentum dissipation is turned off in Ma than in Mbe & Mbb, it is obvious regardless of the physical meaning.

2. How many ensemble members are used to obtain the result? It needs more than only one ensemble member to ensure that the results are robust and meaningful. Also, in general, the MJO prediction skill has been defined by the bivariate correlation coefficient (e.g., Gottchart et al. 2010; Rashid et al. 2011; Kim et al. 2014; Lim et al. 2018). It should be noted that the prediction skill used in this study is not the same as those in the previous studies. I recommend using the word "pattern correlation analysis" rather than the word "correlation analysis."

3. L231-232: Why did you emphasize the clear-sky area? In my eyes, the stronger convection centered over 150E for 24 January – 30 January is already shown, resulting in the difference in the clear-sky area. It might affect the clear-sky area (related to the suppressed convection).

4. L246-249: Why does the convective friction suddenly influence the prediction only over the last days, not over the whole forecast lead time? In my opinion, it seems to be related to the above sampling issue and does not have a physical meaning.

5. Section 3.5 (L321-340): How does the cyclone propagate from 80N to 30N directly? Based on your explanation, the cyclonic circulation comes out from the polar vortex and straightly propagates from 80N to 30N in a limited and fixed longitudinal band (20E-60E). Is it possible? Please show the longitude-latitude section of the 150-hPa stream function in each time step. It will be more helpful for examined the intrusion of a Rossby wave train. Also, why is the intrusion shown only in the longitude band (20E-60E)? Please explain it in detail.

Minor comments

1. L230-231: Could you add a line from the CF into not only the Figure 4 but also all other hovmoller diagrams for comparison?

2. L235–236: The better propagation in Figs. 4b, c than in Fig. 4a might be related to the weaker amplitude of enhanced convection (Seo and Kumar 2009). If you explain the result kindly, it would be more helpful for readers.

3. Figure 6b: Please check the label in Y-axis.

4. Figure number in Figure: Figure 1d → 1c, Figure 3d → 3c.

5. L361: What is "the radiation of an anticyclonic Rossby-wave train"?

6. L16: "to be forecast" → "to be forecasted"

7. L185: Wang et al. 2018 → Wang et al. 2019?
* * *

---

## Author Comment (AC1) · 13 Mar 2020

Thank you for your comments posted on 3 March 2020.

I respond to those as follows:

General Introductory Remarks:

An explorative nature of the present manuscript is emphasized. As well summarized as items 1) and 2) by the present Reviewer, the scientific questions of this study are well posed. We strongly believe that presented results are rich in implications. However, as the present Reviewer suggests, we are short of developing full interpretations of

the results. It is a major reason that we decided to submit the present manuscript to ACP, thus by going through the discussion session, we can obtain various useful feedback. Regardless of the amount of feedback we may receive, we also believe that the materials presented herein strongly invite for theoretical interpretations, that must further be developed. Development of such theories must be a common effort of the community by placing these materials in public domain. This is the main reason that we believe that the present materials are worthwhile to publish in the present form. As stated in the current version, we believe it important to present those details of the model sensitivities so that the theoretical community will be aware of the real issues of the operational MJO modelling.

Major Comments:

1. We believe the difference of different simulations are already carefully described. If the Reviewer believes that further details are required, please be specific. The physical details of IFS are available on Web, and the Web address will be provided in the final version. On more specifics,

i) The momentum dissipation is expected to suppress a "free dynamics", thus we expect that the MJO would also be enhanced by turning it off, if it is described as a free dynamics to some extent. This very basic point will be more explicitly stated in the final version.

ii) However, the most fascinating aspect of the result is that the change of MJO behaviour is hardly monotonous by simply turning off various moment-dissipation terms. In other words, the role of momentum dissipation is highly nonlinear in the MJO dynamics, as already suggested in the manuscript. This point will also be more clearly stated in the final version .

iii) I agree that convection in Fig. 4(a) is stronger than that in Fis. 4(b) and (c). This point will be explicitly stated in the final version.

2. No ensemble run is considered in the present study. Every run is initiated with an initial condition for the operational standard run. Thus, the model is initialized by the most-likely state, and the resulting forecast is also the most-likely evolution under a given physical setting. We do not understand why an ensemble is important for the present purpose, because the most-likely evolution is the main result that we want to know, though ensemble information may provide supplementary information.

In the final version, the term "correlation analysis" will be replaced by "pattern correlation analysis", as suggested.

3. In Fig. 4(a), the most remarkable improvement is the clear-sky area behind MJO. A mechanism for this change is hard to identify, though the present Reviewer may like to speculate. Nevertheless, it does not prevent us from pointing out this most remarkable improvement. Convection associated with MJO is too strong with this setting, and this is hardly considered an improvement as dramatic as the clear sky.

4. As already remarked in response to the item 1, the effect of turning off a physical process is hardly linear, but the MJO evolves in nonlinear manner in response. This is just an example of such a nonlinear that turning off the convective friction leads to a sudden deterioration of the forecast towards the end of the forecast period. Data sampling may only artificially remove those nonlinearities that are actually present.

Minor Comments:

1. Against to what the present Reviewer suggests, there is no line for the CF in Fig. 4. In any case, such a line must be drawn somehow in a subjective manner, because the MJO is hardly a simple linear propagation process. In our opinion, it rather hinders us form more objectively see a change of the forecast by a change of physics.

2. This is a very good speculation to make: indeed, if the MJO is a free wave to a good extent, too strong convection will hinder a proper propagation tendency. This remark will be added in the final version.

3. The y-axis here is correct. Note that the extended Mbb case is run for only 30~days, as stated in Sec. 2.3. The figure caption will be modified for a better clarity in the final version .

4. Thank you for pointing us errors in figures. These errors will be corrected in the final version.

5. "Emission of an anticyclonic Rossby-wave train from the Eastern Pacific towards higher latitudes"

6. The verb "forecast" can be either "forecast" or "forecasted" in past participle form. According to whatis.techtarget.com: Although both are used, forecast is the preferred form.

7. Thank you for picking up a typo: "Wang et al. 2018" will be corrected to "Wang et al. 2019" in final form.

Finally, thank you for suggesting various additional references. However, it would be much helpful, if you could provide us full references, rather than just years and authors' names.
* * *

---

## Referee Comment (RC2) · Anonymous Referee #2 · 17 Mar 2020

By using the ECMWF IFS Cy43r3, this study investigates the sensitivity of MJO prediction to various configurations of physics in the model, with an assumption that the MJO is a free nonlinear Rossby wave. Sensitivity tests include turning off the momentum dissipation and diabatic heating. Authors conclude that the reduction of momentum dissipation improves the MJO prediction but leads to a weaker MJO overall. My overall impression is that, there may be many interesting results, especially that can help the world-best MJO forecast model to be even better. Therefore, I agree with the authors that (from the reply to the reviewer's comment) this is a significant study. However, as the other reviewer pointed out already, it is very hard to follow their argument, and I think this is mainly because of the presentation of the results (low-quality figures) and too

complicated results due to various sensitivity tests performed. I don't think the readers will get to the key point of this study with the present form. Here are some suggestions to improve the manuscript. By doing that, the results would be more convincing and easier to review.

1) Presentation of Figures: There are many aspects of atmospheric convection and circulation discussed with showing only the hovmuller plot, which averaged out many detailed structures of what the author wants to explain. For example, I am not sure how I can see the "anticyclonic vortex pair symmetric to the equator" (Line 132) with the 15S-15N averaged plots. So, additional figures besides Hovemuller will help.

2) Hovmuller plots can be further improved this way: i) only show one shading bar on the bottom of each figure if they present the same interval. ii) shorten the title of each figure (for example, "Hovmuller of stream function 150 hPa (20.0N-0.00N)" doesn't need to be repeated in each figure. Or just simply say "SF150: Analysis", "SF150: CF" or something like this. iii) There are also some typos in figures: For example, two '(b)' in Figure 3 which detracts me from reading. iv) In all Hovmuller plots, there is a white shading but does not present in the color bar. What is it?

3) Results: I'd suggest to only pick several sensitivity experiments, focus on them, and discuss them thoroughly with strongly connecting the results with the existing theories.

4) MJO event selection: The selection of this specific event needs to be better justified. I don't understand why 'a low-skill event' is selected. Also, as the other reviewer pointed out, comparing a standard index (RMM index) would be useful.

---

## Editor Comment (EC1) · Peter Haynes (Editor) · 31 Jul 2020

The two referees' reports are strongly critical of this paper primarily on the basis of the lack of clarity of the description of the approach taken, of the scientific questions being addressed and of the logical arguments that are required to use the simulation results to answer the scientific questions.

The authors made a reply in the on-line discussion to one of the referee reports. (The second report was posted two weeks later.) This reply began by emphasising the explorative nature of the paper and later gave a statement of the rationale for the paper – that the set of simulations and the results from them was a potentially valuable resource for researchers. That is a noble rationale, but it does require the authors to think clearly about how their results – which, I think it is fair to say, do not lead to any straightforward constraints on theoretical interpretation of the MJO and associated processes and mechanisms, can be presented in a way that will be genuinely useful to other scientists working on this problem.

Perhaps the paper needs a frank statement that no simple theoretical interpretation of the results has been possible. The statement is made in the abstract that 'A motivation behind this study is to explore a possibility of interpreting the MJO as a nonlinear free wave under active interactions with Rossby waves from and to higher latitudes'. Perhaps that is confusing in itself – because the reader may think that some progress on this must have been achieved – when in fact it has not.

Looking at the Discussion section of the paper it is very difficult to extract any firm conclusions. Do the first two paragraphs leave open the possibility that the free nonlinear Rossby-wave theory is viable? Or not? Does a later paragraph support the idea that taking account of interactions with extratropical waves may improve predictability? Or not? There are some very general statements about the subtleties of switching different processes on or off and interpreting the results. But it didn't require the results presented from this paper to establish that.

The reply also asked the referee for more specific details of the extra information requested in their report. My comment on this is that both referees have found it very difficult to follow the arguments presented in the paper. When I am in this situation myself as a referee I sometimes feel that I can give a rather specific set of instructions to the authors. ('If you clarify points A, B , C and D then the paper will be suitable for publication.') But other times it is not at all straightforward to do that – in such a case all one can do is report that one has found the paper unclear in various aspects and then put the onus on the authors to resolve this. I think that in this case the authors simply have to look at the paper again and consider how they might get over their arguments more clearly. [The reply requests further information on references – it was not difficult

to find these – see list at end of this comment.]

My overall view as Editor at this stage is that very significant revision will be required in order to bring this paper to a form suitable for publication in ACP. The MJO is of course a very important and challenging topic and the kinds of experiments that the authors report are not straightforward (and for many researchers are out of reach). Therefore the authors' wish that their results should be on record so to allow others to benefit from them is not an unreasonable justification for publication. But my impression from the two referees' reports is that the paper in its current state is not useful – there is little likely benefit to other scientists wanting to make progress in this field. The authors have to consider carefully what information and accompanying discussion is needed in the paper to improve that situation.

The authors have asked for extensions of the deadline for revision of their paper because of disruption to their work due to the covid-19 pandemic. My advice (offered late, but with the partial excuse of the same disruption) is, given the substantial revision that will be required, the authors do not proceed with revision of the paper, but instead spend some time considering carefully how their work can be presented in a way that will be genuinely useful to others and after that, if appropriate, make a new submission.

REFERENCES

Gottschalck, J., and Coauthors, 2010: A framework for assessing operational Madden–Julian oscillation forecasts: A CLIVAR MJO Working Group project. Bull. Amer. Meteor. Soc., 91, 1247–1258, https://doi.org/10.1175/2010BAMS2816.1.

Kim, H.-M, P. J. Webster, V. E. Toma, and D. Kim, 2014: Predictability and prediction skill of the MJO in two operational forecasting systems. J. Climate, 27, 5364–5378, https://doi.org/10.1175/ JCLI-D-13-00480.1.

Lim, Y., Son, S.-W., Marshall, A. G., Hendon, H. H., & Seo, K.-H., 2019: Influence of

the QBO on MJO prediction skill in the subseasonal- to-seasonal prediction models. Climate Dynamics, 53, 1–15.

Rashid, H. A., H. H. Hendon, M. C. Wheeler, and O. Alves, 2011: Prediction of the Madden–Julian oscillation with the POAMA dynamical prediction system. Climate Dyn., 36, 649–661, https://doi.org/10.1007/s00382-010-0754-x.

Seo, K., W. Wang, J. Gottschalck, Q. Zhang, J. E. Schemm, W. R. Higgins, and A. Kumar, 2009: Evaluation of MJO Forecast Skill from Several Statistical and Dynamical Forecast Models. J. Climate, 22, 2372–2388, https://doi.org/10.1175/2008JCLI2421.1.

---

## Author Comment (AC2) · 24 Aug 2020

We much appreciate the comments posted on 17 March 2020 by the present Reviewer. Our apologies for a delay of a response from our side, mostly due to a *confinement* of the first author during the epidemic.

Indeed, the present Reviewer provides us with a very good summary of the present study. We also much appreciate a positive evaluation, stating that "there may be many interesting results, especially that can help the world-best MJO forecast model to be even better. Therefore, I agree with the authors that (from the reply to the reviewer's comment) this is a significant study." However, as the case with the first reviewer, the

present Reviewer remarks that "it is very hard to follow" mostly due to "too complicated results".

Yes, the results are "very complicated" with very different behaviour sensitively depending on the choice of physics to be turned off. These are results that we even did not expect when we started this project. However, we are afraid that we must best present these complicated results as they are, because these are what we get. By reading through the original manuscript carefully, we realise that the main problem was in presenting our original motivation of the study as if the purpose of the paper itself. The real purpose here is to report these complicated results, which do not give any clear-cut interpretations in terms of the nonlinear free–Rossby wave dynamics as we originally envisioned. The original manuscript was hard to read, because we presented the results without warning the readers properly. In the revised manuscript, we will make this point as clear. Furthermore, more interpretational remarks will be inserted into the analysis section so that readers may not be get lost in details.

We agree that, as a reviewer may react, it is very usual just to report all those details of model sensitivities as a scientific report. However, the first author, especially, points out that the very fact of never reporting those modelling sensitivities is a core reason for slow progress of global models, without much useful inputs from theoretical studies. A commentary to Nature by Metha (2019) makes the merit on this type of publications clear.

For example, as already suggested in the original introduction, there are extensive studies in theoretical literature about whether the friction contributes to the MJO dynamics positively or negatively. However, all these studies are based on a rather simple Rayleigh–friction formulation with a dichotomy of with or without friction. A very important message from the present study is that the effects of friction is hardly such a simple dichotomy. Rather the performance of MJO prediction sensitively depends on the choice of the exact friction term. This very fact is something to be reported to the theoretical community so that theoreticians can more positively contribute to un-

derstand these "complex" behaviours of MJO within global models.

Another important message to convey from the present study is a difficulty of emulating the free dynamics within an operational global model: if we totally turn off the dissipation terms as well as diabatic heating, as attempted in this study, the basic climatology, that is required to support a free nonlinear–wave dynamics, is also destroyed as a result, thus an expected free dynamics is no longer simulated. The significance of these lessons from the present study will be more clearly highlighted in revision.

As for more specific issues:

1) Presentation of Figures: We decided to focus on the Hovmuller plot, because we find it the most succinct manner of presenting the MJO behaviour both in terms of convection and vorticity (rotational flows). Under this configuration, "anticyclonic vortex pair symmetric to the equator" appears as a positive anomaly in a Hovmuller plot, as already remarked in Sec. 2.1. To make a point clearer, we will add a phrase "over the Indian Ocean" in revision. We will also change the phrase itself as "anticyclonic activity", because it is true that by Hovmuller plot only, it is not possible to tell, whether this is a vortex pair or not. A reader would be able to identify a development of a positive stream–function anomaly along a MJO propagation easily in this manner. On the other hand, the role of the vortex pair in the theory will be explicitly remarked in revision for better clarity.

2) Improvement of Hovmuller plots:

i) Though it would be possible to remove some redundant color bars from figures, presentation of figures would become less coherent as a result. For this reason, we opt not to perform this change.

ii) In Revision, "Hovmoeller of" will be removed from all the figure headings as suggested. Similar simplifications of the figure headings will also be applied to Figure 2.

iii) There was a problem with sub-labels in the original Figure 3. This will be corrected

in final version.

iv) The values beyond a range of colour code is not shaded. This fact will be remarked in the revised caption of Figure 1 in such manner that the remark also applies to all the subsequent figures.

3) Results: For reducing the amount of results to be presented, Sec. 3.5.2 will be removed in revision, because it does not offer much. Nevertheless, the main message to be conveyed by the present paper is the very fact that none of the existing theories appears to explain the identified complex sensitivities. In this very respect, a number of cases is important to explicitly indicate a complex response of the model by selectively turning off the physics.

4) MJO event selection: A 'low–skill event' is selected in the present study, because by definition, it is more challenging to forecast. As shown in Fig. 3, the performance of controls runs is rather poor. Thus, the question is: how can we improve it? As reported herein, we have certain successes. However, the change of the model performance is not quite consistent in terms of change of contribution of friction. We believe that the latter is more important to emphasise rather than reporting some limited successes, which are only superficially good news.

By following a suggestion of the present Reviewer, some diagnostics based on RMM indices will be added in revision (new Fig. 3).

**references**:

Mehta, D., 2019; Highlight negative results to improve science. *Nature*.

https://www.nature.com/articles/d41586-019-02960-3
* * *

---

## Author Response (AR1)

**General Remarks**

The manuscript has been thoroughly revised by fully taking into account the comments by the Editor, Peter Haynes, as well as by the two anonymous reviewers. Our responses originally posted to the ACPD Web site are appended below with only minor edit, because the revision follows our responses already provided. The line numbers are added systematically in the following, including in those individual responses, to refer exactly where these suggested modifications are made.

A major criticism common to all the comments received concerns readability. To improve the manuscript, a major re–structuring of the text has been performed at various places, most notably in the introduction and the final section. They should make the motivation and the goals clearer, while the main results of the work are presented in a more logical, sequential manner.

An outline of the analysis section (Sec. 3) has been added in the end of the first subsection (Sec. 3.1) so that readers can follow these rather detailed presentation of the complex results with an overall structure of the presentation in mind. This should substantially reduce the risk of readers getting lost in the details. Some general remarks are also inserted (L269, L297, L308–309, L315–317) throughout this section as a further guidance for readers.

In an effort to remove rather secondary details, Sec. 3.5.2 and the associated two figures (Figs. 9, 10) are removed in the revised manuscript.

For improving the overall clarity of the presentation, two tables are added that summarise the experiment categories and the individual forecast cases.

By following a request of Reviewer #2, some examples of the RMM analysis are also added in revision (new Fig. 3 with the original figures re–numbered).

The main weakness of the original manuscript was a lack of conclusions. In revision, both in the abstract and in the final section, we point out the following conclusions:

- A difficulty of emulating free dynamics in a global model without destroying a realistic background state necessary for supporting a free dynamics (L9–10, L399–404)
- Strong sensitivities of the MJO simulation on physics rather other than exclusively on convection parameterizations (L10–11, L449–453)
- The need for active contributions of theoretical studies that are more closely tied to the specific realisation of physics in global operational models (L6–9, L82–87, L454–461) See also L94–96 in the introduction.

In the final section, the following points are further emphasized:

- Strong sensitivity of the MJO predictability on the initial condition (L421–429)
- Active interactions of MJOs with higher-latitude Rossby-wave activities (L441-448)

We hope that our revision is satisfactory for publication in ACP.

**Individual Responses**

**Reply to Editor**

We would like to thank the Editor for his considerate comments. We apologise again for the delay, partially this was due to Covid-19 as one of the authors had difficulties to access the relevant infrastructure and data. Having said this, we much appreciate the opportunity provided by the Editor to respond in full. We understand that it is not easy to give specific recommendations as an editor, when the main issue is in the presentation of difficult and unexpected results. We *do* believe that the results should be recorded to illustrate, for example, the contrast of friction formulations in operational models with those in idealised studies of the MJO. The latter are often based on a rather simple Rayleigh–friction formulation, taking a dichotomy of with or without. Unfortunately, existing theoretical models are too limited to explain corresponding MJO sensitivities simulated by operational models.

We agree that no simple theoretical interpretation of the results are possible from the present study. Originally, we started to explore a possibility of interpreting the MJO as a nonlinear free wave under active interactions with Rossby waves from and to higher latitudes. A main strategy has been to remove the constraints to the free dynamics in an operational model by selectively turning off the tendencies of different physical parametrisations (L37–40, L68–70, L75–79, L87–89).

In spite of a substantial number of sensitivity experiments performed, it turns out to be difficult to draw firm conclusions. However, such a work should not simply be considered a failure. Here, we disagree with the Editor's comment that "the current state (of manuscript) is not useful", if what it means is a lack of positive results. Notably, in a recent comment in the journal Nature, Mehta (2019) argues why a negative result is crucial for a healthy progress of science.

It would be important to emphasise that our methodology is sound, and we have set out with a clear hypothesis as stated in the manuscript. More specifically, individual sensitivities of momentum diffusion are examined, with a hope of distilling specific impacts that either deteriorate or improve MJO forecasts. As it turns out, such an investigation is difficult, because other processes, that are not eliminated, compensate with a nonlinear response. We agree with the assessment that the results are complicated. However, it is unethical to simplify what we actually obtained. We further agree with the Editor and the Reviewers about the (lack of) presentation style of these complicated results. We have revised the manuscript to better present the unexpected complexity of the results (*e.g.*, L213–225). We also clearly state in revision that we do not find any clear-cut interpretations in terms of the nonlinear free-Rossby wave dynamics as we originally envisioned (L75–81, L389–404). Nevertheless, this is an important negative finding, that should inspire further experimental studies while avoiding repeating the same mistakes made here (L82–87, L454–461).

As recommended by Mehta (2019), the present manuscript will become a showcase that established researchers with a good background on the MJO fail to prove their hypothesis. It will further send a strong message to younger and aspiring scientists bombarded with success stories. It is our view that the Reviewers and the Editor read the present manuscript with 'success' in mind.

More specifically, in revision, we have realised that it is difficult to extract any firm conclusions for readers from the original manuscript for two related reasons. First, the basic nature of the present study is exploratory (L75). The main goal is an extensive sensitivity study of MJO forecasts on physics, that call for theoretical studies more closely tied to the actual physics of operational models (L75–86). Urgent needs for such a new type of theoretical studies are more explicitly emphasised in revision (L82–87, L449–461). Second, we have failed to state this actual main goal of the work in the original manuscript. The motivation of the study to investigate a possibility of interpreting MJO as a nonlinear free wave in operational models is wrongly stated as a main purpose (L2–3, L389–391). This has been corrected in revision so that readers will be better guided through the revised manuscript (L219–222, L391–397).

We still personally believe that the free nonlinear Rossby-wave theory remains a viable idea. However, clearly, we have failed to obtain any firm support to this theory by the present sensitivity study (L396–397). It simply demonstrates how hard it is to emulate free dynamics within a global forecast model without deteriorating the basic state of the model that so crucially depends on these physical parametrisations. This point has already been made in the original manuscript. However, we have failed to extend its implications (L399–420).

In contrast, we have obtained firm evidence for interactions of the MJO with extratropical waves by the present sensitivity study: the behaviour of the model is relatively insensitive to the choice of physics in representing this aspect of the MJO dynamics. This very point, that was failed to be remarked in the original manuscript, has been clearly be pointed out in revision (L223–225, L441–448).

The Editor suggests that finding sensitivities themselves do not constitute anything original. However, we disagree on this point in the context of MJO studies: these studies are strongly driven by a paradigm of MJO driven by convection, thus almost any global modelling studies of the MJO are also exclusively focused on sensitivities to convection parametrisation. A recent paper by Pilon et al (2016) and Jiang et al (2020) are a good example. The originality of the present paper is to explicitly point out that MJO forecasts do not sensitively depend on convection parametrisations only but also on other physics, especially the momentum dissipation processes (L20–36, L82–87, L449–453). Probably, pointing out this very simple fact is already a very important contribution of the present work. Unfortunately, we had failed to emphasise such a basic point in the original manuscript.

**Respond to the Reviewer 1**

Thank you for your comments posted on 3 March 2020.

I respond to those as follows:

**General Introductory Remarks:**

An explorative nature of the present manuscript is emphasised (L75). As well summarized as items 1) and 2) by the present Reviewer, the scientific questions of this study are well posed. We strongly believe that presented results are rich in implications (L20–36, L82– 87, L449–453). However, as the present Reviewer suggests, we are short of developing full interpretations of the results. It is a major reason that we decided to submit the present manuscript to ACP, thus by going through the discussion session, we can obtain various useful feedback. Regardless of the amount of feedback we may receive, we also believe that the materials presented herein strongly invite for theoretical interpretations, that must further be developed (L6–9, L82–87, L449–461). Development of such theories must be a common effort of the community by placing these materials in public domain. This is the main reason that we believe that the present materials are worthwhile to publish in the present form. As stated in the current version, we believe it important to present those details of the model sensitivities so that the theoretical community will be aware of the real issues of the operational MJO modelling.

**Major Comments:**

1. We believe the difference of different simulations are already carefully described. If the Reviewer believes that further details are required, please be specific. The physical details of IFS are available on Web, and the Web address has been provided in the final version (L133–134). On more specifics,

i) The momentum dissipation is expected to suppress a "free dynamics", thus we expect that the MJO would also be enhanced by turning it off, if it is described as a free dynamics to some extent. This very basic point has been more explicitly stated in the final version (L37–40).

ii) However, the most fascinating aspect of the result is that the change of MJO behaviour is hardly monotonous by simply turning off various moment-dissipation terms (L4–6, L89– 90). In other words, the role of momentum dissipation is highly nonlinear in the MJO dynamics, as already suggested in the manuscript. This point has also been more clearly stated in the final version. [The word "highly nonlinear" was not used in revision, because it sounds rather too strong, but the nonlinear response of the model to physics is much emphasised in revision (L87–88, L219–222, L392–394).]

iii) I agree that convection in Fig. 4(a) is stronger than that in Figs. 4(b) and (c). This point has been explicitly stated in the final version (L269–270, L276–277).

2. No ensemble run is considered in the present study. Every run is initiated with an initial condition for the operational standard run. Thus, the model is initialised by the most-likely state, and the resulting forecast is also the most-likely evolution under a given physical setting. We do not understand why an ensemble is important for the present purpose, because the most-likely evolution is the main result that we want to know, though ensemble information may provide supplementary information.

In the final version, the term "correlation analysis" has been replaced by "pattern correlation analysis", as suggested.

3. In Fig. 4(a), the most remarkable improvement is the clear-sky area behind MJO. A mechanism for this change is hard to identify, though the present Reviewer may like to speculate. Nevertheless, it does not prevent us from pointing out this most remarkable improvement. Convection associated with MJO is too strong with this setting (L269–270), and this is hardly considered an improvement as dramatic as the clear sky.

4. As already remarked in response to the item 1, the effect of turning off a physical process is hardly linear, but the MJO evolves in nonlinear manner in response. This is just

an example of such a nonlinear that turning off the convective friction leads to a sudden deterioration of the forecast towards the end of the forecast period. Data sampling may only artificially remove those nonlinearities that are actually present.

**Minor Comments:**

1. Against to what the present Reviewer suggests, there is no line for the CF in Fig. 4 (Fig. 5 after revision). In any case, such a line must be drawn somehow in a subjective manner, because the MJO is hardly a simple linear propagation process. In our opinion, it rather hinders us form more objectively see a change of the forecast by a change of physics (*cf.*, L216–222, L430–440).

2. This is a very good speculation to make: indeed, if the MJO is a free wave to a good extent, too strong convection will hinder a proper propagation tendency. This remark has been added in the final version (L276–277).

3. [Fig. 7(b) in revison] The y-axis here is correct. Note that the extended Mbb case is run for only 30 days, as stated in Sec. 2.3. The figure caption has been modified for a better clarity in the final version .

4. Thank you for pointing us errors in figures. These errors have been corrected in the final version.

5. "Emission of an anticyclonic Rossby-wave train from the Eastern Pacific towards higher latitudes" [The whole subsection containing this sentence has been removed in revision]

6. The verb "forecast" can be either "forecast" or "forecasted" in past participle form. According to whatis.techtarget.com: Although both are used, forecast is the preferred form.

7. Thank you for picking up a typo: "Wang et al. 2018" has been corrected to "Wang et al. 2019" in final form.

**Respond to the Reviewer 2**

We much appreciate the comments posted on 17 March 2020 by the present Reviewer. Our apologies for a delay of a response from our side, mostly due to a *confinement* of the first author during the epidemic.

Indeed, the present Reviewer provides us with a very good summary of the present study. We also much appreciate a positive evaluation, stating that "there may be many interesting results, especially that can help the world-best MJO forecast model to be even better. Therefore, I agree with the authors that (from the reply to the reviewer's comment) this is a significant study." However, as the case with the first reviewer, the present Reviewer remarks that "it is very hard to follow" mostly due to "too complicated results".

Yes, the results are "very complicated" with very different behaviour sensitively depending on the choice of physics to be turned off. These are results that we even did not expect when we started this project. However, we are afraid that we must best present these complicated results as they are, because these are what we get. By reading through the original manuscript carefully, we realise that the main problem was in presenting our original motivation of the study as if the purpose of the paper itself. The real purpose here is to report these complicated results, which do not give any clear-cut interpretations in terms of the nonlinear free–Rossby wave dynamics as we originally envisioned (L75– 87). The original manuscript was hard to read, because we presented the results without warning the readers properly. In the revised manuscript, we have made this point as clear (*e.g.*, L213–225). Furthermore, more interpretational remarks have been inserted into the analysis section so that readers may not be get lost in details (L269, L297, L308–309, L315–317).

We agree that, as a reviewer may react, it is very usual just to report all those details of model sensitivities as a scientific report. However, the first author, especially, points out that the very fact of never reporting those modelling sensitivities is a core reason for slow progress of global models, without much useful inputs from theoretical studies (L7– 11, L75–86, L183–186, L397–398, L454–461). A commentary to Nature by Metha (2019) makes the merit on this type of publications clear.

For example, as already suggested in the original introduction, there are extensive studies in theoretical literature about whether the friction contributes to the MJO dynamics positively or negatively (L39–43). However, all these studies are based on a rather simple Rayleigh–friction formulation with a dichotomy of with or without friction (L84–85, L459–461). A very important message from the present study is that the effects of friction is hardly such a simple dichotomy. Rather the performance of MJO prediction sensitively depends on the choice of the exact friction term. This very fact is something to be reported to the theoretical community so that theoreticians can more positively contribute to understand these "complex" behaviours of MJO within global models (L80–87, L454–461).

Another important message to convey from the present study is a difficulty of emulating the free dynamics within an operational global model (L9–10, L399–404): if we totally turn off the dissipation terms as well as diabatic heating, as attempted in this study, the basic

climatology, that is required to support a free nonlinear-wave dynamics, is also destroyed as a result, thus an expected free dynamics is no longer simulated. The significance of these lessons from the present study has been more clearly highlighted in revision (L399–404).

As for more specific issues:

1) Presentation of Figures: We decided to focus on the Hovmuller plot, because we find it the most succinct manner of presenting the MJO behaviour both in terms of convection and vorticity (rotational flows: L181–186). Under this configuration, "anticyclonic vortex pair symmetric to the equator" appears as a positive anomaly in a Hovmuller plot, as already remarked in Sec. 2.1. To make a point clearer, we have added a phrase "over the Indian Ocean" in revision. We will also change the phrase itself as "anticyclonic activity", because it is true that by Hovmuller plot only, it is not possible to tell, whether this is a vortex pair or not (L144). A reader would be able to identify a development of a positive stream–function anomaly along a MJO propagation easily in this manner.

2) Improvement of Hovmuller plots:

i) Though it would be possible to remove some redundant color bars from figures, presentation of figures would become less coherent as a result. For this reason, we opt not to perform this change.

ii) In Revision, "Hovmoeller of" has been removed from all the figure headings as suggested. Similar simplifications of the figure headings have also been applied to Figure 2.

iii) There was a problem with sub-labels in the original Figure 3. This has been corrected in final version.

iv) The values beyond a range of colour code is not shaded. This fact has been remarked in the revised caption of Figure 1 in such manner that the remark also applies to all the subsequent figures.

3) Results: For reducing the amount of results to be presented, Sec. 3.5.2 has been removed in revision, because it does not offer much. Nevertheless, the main message to be conveyed by the present paper is the very fact that none of the existing theories appears to explain the identified complex sensitivities. In this very respect, a number of cases is important to explicitly indicate a complex response of the model by selectively turning off the physics.

4) MJO event selection: A 'low-skill event' is selected in the present study, because by definition, it is more challenging to forecast (L139–142). As shown in Fig. 3, the performance of controls runs is rather poor. Thus, the question is: how can we improve it? As reported herein, we have certain successes. However, the change of the model performance is not quite consistent in terms of change of contribution of friction (L216–221, L392–394). We believe that the latter is more important to emphasise rather than reporting some limited successes, which are only superficially good news (L222–223, L454–461).

By following a suggestion of the present Reviewer, some diagnostics based on RMM indices have been added in revision (new Fig. 3).

**references:**

Mehta, D., 2019; Highlight negative results to improve science. *Nature*. https://www.nature.com/articles/d41586-019-02960-3

Jiang, X., E. Maloney, and H. Su, 2020: Large-scale controls of propagation of the Madden-Julian Oscillation. *Clim. Atmos Sci*, **3**. https://doi.org/10.1038/s41612-020-00134-x

**Sensitivities of the MJO Forecasts on Configurations of Physics in the ECMWF Global Model**

Jun–Ichi Yano1 and Nils P. Wedi2

1CNRM, Météo-France and CNRS, UMR 3589, 31057 Toulouse Cedex, France 2European Center for Medium–Range Weather Forecast, Reading, UK

Correspondence: Jun-Ichi Yano (jiy.gfder@gmail.com)

**Abstract.** Sensitivities of MJO forecasts to various different configurations of physics are examined with the ECMWF global model, IFS. The motivation of the study is to simulate the MJO as a nonlinear free wave under active interactions with Rossby waves from and to higher latitudes. To emulate free dynamics in IFS, various momentum dissipation terms ("friction") as well as diabatic heating are selectively turned off over the tropics for the range of the latitudes 20S-20N. The reduction of friction

- 5 tends to improve the MJO forecasts, but hardly in any additive manner. A change of the forecast performance rather sensitively depends on the type of friction turned off. The behaviour is in contrast to many theoretical studies based on a rather simple Rayleigh–friction formulation under a dichotomy of with or without. By reporting the details of those physical sensitivities on the MJO forecast, the present study suggests a need for theoretical investigations that much more closely follow the actual operational formulations of physics. An important lesson to learn from the study is an inherent difficulty to emulate 
[revised manuscript text omitted]

As another summary for the forecast performances, Fig. 3 present RMM analyses for some selective cases. Here, (a) and (b), respectively, show the evolution trajectory of the analysis data on the RMM phase space over the standard and extended forecast periods. Evolution of the MJO is represented by a counter-clockwise movement of a trajectory in this phase space, with an initial point marked by a red circle, as seen in both frames. Note that although the extended forecast period contains the standard forecast period as a part, the two trajectories for the ERA5 analysis do not match exactly over the same period

8

due to the different definitions of the anomaly used (defined relative to an average over a selected forecast period). These two trajectory patterns are to be compared with those of sensitivity experiments and control forecasts as a verification.

The remainder of this section proceeds as follows: morphological behaviours of the control forecasts are carefully described in the next subsection (Sec. 3.2), because they provide baselines for interpreting subsequent runs turning–off selected physics.

- 215 The following two subsections (Secs. 3.3 and 3.4) look for improvements of MJO forecasts by removing momentum dissipations as well as diabatic heating effects, as would be expected from the free nonlinear Rossby–wave theory. As it turns out the performance of the MJO forecasts does not depend on these choices of physics in any consistent manner: less momentum friction does not necessarily lead to a further improved MJO forecast, but the skill and MJO propagation sensitively depends on the type of dissipation turned off. Effects are hardly additive, either, but clearly nonlinear interactions are going on between
- 220 the physics. Thus, against the original motivation stated in the introduction, the main purpose of these two subsections becomes a report of these forecast sensitivities in more detail. Careful descriptions will also reveal that improvements of the MJO forecast is hardly a monotonic measure: certain aspects are improved, but often associated with deterioration of other aspects. Sec. 3.5 focuses on the model performance on simulating interactions between the MJO and higher–latitude Rossby–wave activities. Here, we rather find a consistent tendency that the model simulates those interactions features identified in the analysis relatively well, although some sensitivities inevitably emerge.
  - 3.2 Control Forecasts (CFs)

This subsection first establishes basic behaviours of the control forecasts (CFs), because they are the base lines for defining a change in forecasts by turning off certain physical processes.

[revised manuscript text omitted]

---

## Referee Report (RR1)

Overall opinion

This study tries to answer the scientific questions related to 1) the possibility of interpreting the MJO as a free nonlinear Rossby wave through turning off the diffusion terms and 2) the high latitudinal impact on the MJO initiation process. In the revised manuscript, I think the scientific questions become clearer, and the explanations become more detailed. However, the main results are not sufficient to support two scientific questions (L55-58). The results for the first question and its related explanations are not well summarized. It is still hard to figure out the main points. The results for the second question are not enough to argue the role of extratropical forcing. More specific comments are written below.

Major comments

1. L143-147 "From a dynamical point of view, this is before the anticyclonic activity begins to develop over the Indian Ocean (Fig. 1(d)). Thus the key forecast question is whether the model can predict the onset of this activity". It isn't clear. The sentence is not logically connected with the first two sentences in L143-145.

2. L279-281: I cannot figure out the author's point in this sentence "the emission of the Rossby wave energy from west during 22-28 January is suggested as a major source for initiating the anticyclonic signal associated with the MJO by the time-longitude plots (Fig. 6(a) for Ma)". Do the authors think that Ma's better simulation is from the Rossby wave energy? I could not find any evidence about the anticyclonic signal is the major source.

3. L316-317: In 3.3.1-3.3.2, the authors show that the dissipation terms and convective frictions affect the MJO simulation. However, does the sentence in L316-317 mean the results are dependent on the integration time?

4. L297: Based on the results from Mbc and Mbs, how did the authors get those interpretations? Please discuss more in detail.

5. L337-343: The authors argue that the pattern correlation is recovered by the westward propagating Rossby wave. In my opinion, "recovery" is not right expression. The expression exaggerates the results, and the high pattern correlation is a coincidence. I think this result cannot support the impact of the Rossby waves on the MJO simulation. Please carefully discuss it.

6. Second scientific question and Figure 9: I'm suspicious that the Rossby-wave train is really important in this event. If the Rossby-wave train is important, please show the longitude-latitude plot with time integration. The Rossby-wave train does not directly propagate into the lower latitude with no change in longitudinal location. In this regard, I think Figure 9 is not sufficient to examine the role of the Rossby-wave train. It is also hard to figure out why the results in NQ, QF, and Ma experiments are needed to test the extratropical Rossby wave train.

Minor comments

1. L5: "but hardly in any additive manner": What's meaning?

2. L32: I would recommend to add a short explanation about "free-Rossby wave dynamics

in MJO"

3. L73 & L197: summarised -> summarized

4. L75 & L184: emphasised -> emphasized

5. Y axis in Figs. 1, 4, and 5 are not consistent.  Please revise it.

6. L266: To clarify the information, please remove the sentence. "This subsection discusses this overall aspect. The next subsection focuses more specifically on convective friction."

7. Page 26: Figure.5 => Figure.6

---

## Author Response (AR2)

**General Remarks**

Both the Editor and the Referee #2 find that the manuscript does not yet present the important points well. In the current revision, we have much strengthened the points stated in the abstract and in the final section, while sharpening the narrative and the presentation.

The abstract has been completely restructured, in particular L5–11: "Contrary to the original motivation, emulating a free dynamics with an operational forecast model turns out to be rather difficult, because forecast performance sensitively depends on the specific type of friction turned off. The result suggests a need for theoretical investigations that much more closely follow the actual formulations of model physics: a naive approach with a dichotomy of with or without friction simply fails to elucidate the rich behaviour of complex operational models. The paper further exposes the importance of physical processes other than convection for simulating the MJO in global forecast models."

The discussion section has been revised and a new paragraph has been added (L397–409: see also L88–94 in the introduction).

Especially, the last two sentences of the new paragraph emphasise: "Though we are short of making any definite conclusions from our sensitivity study of the MJO on the momentum dissipation processes, the study suggests a critical importance of examining the physical sensitivities of a phenomenon with more detail rather than simply switching off the entire physical mechanism as has been done in past sensitivity studies." We believe that this is an important message that needs to be widely appreciated in the community.

Please also see the final paragraphs in L458–465.

We would like to thank the Editor and a referee for their comments and suggestions: we have made an extensive re-configuration of the text following the suggestions of the Editor closely.

We hope that the present revision is satisfactory to make the manuscript publishable.

In the following please find detailed responses to the comments and suggestions.

**Individual Responses**

**Reply to Editor:**

We would like to thank the Editor for his review and suggesting "to reconsider the present manuscript after major revisions".

We also much appreciate the comments by the Editor where he agrees with our claim that "a paper containing negative results could be valuable — if the work was clearly presented."

We further interpret an acceptance recommendation by one of the referees, even if without any remarks, to be a positive sign that our previous efforts in revising the manuscript have addressed some of the concerns. Unfortunately, another referee (Referee #2) is still negative, as the Editor also remarks, to which we respond below separately.

The Editor is inclined to agree with the critical referee that there continue to be many aspects of the paper that are unclear; in principle, a paper can be a valuable contribution even if it contains only 'negative' results; nevertheless, the present paper falls short of providing a clear presentation of a set of interesting numerical simulations that will be a genuine help to others who are planning this sort of work.

We have strengthened in response the main points of the paper as summarised both in the abstract and in the final discussion section.

The Editor concludes that the paper may be suitable for publication subject to further revisions. For this step, the Editor asked to consider carefully the comments of the referee as well as the Editor's own, to which we respond below. The Editor suggests to focus much more on a clear description of the simulations and what exactly can be deduced from them — perhaps with some discussion of how the simulations might be modified further to resolve some of the questions that are currently unresolved.

In revision, the abstract has been revised to make the key points clear and revised the discussion section with an additional paragraph (L397–409: see also L88–94 in the introduction). This paragraph explicitly points out an inherent limitation of existing mechanism–denial studies, while emphasising the importance of detailed sensitivity studies of specific physical processes, as in the present work. It would be an important guidance to those who are planning similar work in future.

Though not explicitly stated in the text, we also believe that many of the currently unresolved problems of MJO prediction can be addressed with more extensive, careful sensitivity studies, given more detailed physical constraints. The latter is particularly relevant

in the context of emerging tools based on machine learning for extracting "causal" MJO information from data.

On the other hand, we believe that the description of the simulations is already clearly presented. For this reason, we have mostly addressed the suggestions by the editor by re-configuring the text as described below. A critical future direction is already stated in the previous version (L426–428): to repeat the present sensitivity study by a more elaborate numerical setup (with an explicit ambient state) under development (Kühnlein *et al.* 2019).

Two Particular Concerns:

The following major modifications have been made in response to the Editor's two partic-ular concerns.

Firstly, more recent publications have been added as references. The Editor has kindly suggested several papers, and all of them are cited in revision. The two recent reviews (Zhang *et al.* 2020, Jiang 2020) are added (L25–26). As more specific references, Ma and Kuang (2016) is cited in discussing the difficulty of maintaining a relevant basic state when certain physical processes are turned off (L406–409). We have also emphasised (L425–428) a conceptual difference of our sensitivity study from the so–called mechanism–denial studies (*e.g.*, Kim *et al.* 2011, Ma and Kuang 2016). Furthermore, the question of interactions between the MJO and higher–latitude Rossby–wave activities is better placed in more general modelling contexts by citing Hall *et al.* (2017: L53–55).

In responding to the second major concern, the section plots have been collocated in revision based on the fields (OLR, 250-hPa stream function) and the periods (standard 20–day forecast, 40–day extended forecast, 20–day forecast from 1 Feb.): in all collocations, the first frame is the analysis field, thus it much facilitates comparisons of forecasts with the analysis.

DETAILED COMMENTS

**Abstract**:

L3: The editor suggests here that though the motivation was to seek support for the 'free wave' theory, no support is found. However, the precise conclusion is that perhaps the 'naive' approach taken here does not allow us to draw any conclusions on the theory of the MJO that has motivated this study.

L4-5, 'The reduction in friction tends to improve the MJO forecasts, but hardly in any additive manner.' : This sentence has been modified in revision to: 'The reduction of

friction sometimes improves the MJO forecasts, but without any systematic tendency.'

**Main text**:

L15: 'the pre-existing MJO' has been changed to 'a pre-existing MJO' as suggested

L16: 'especially, in' has been changed to 'especially in' as suggested

L21: "or short 'physics' hereafter" has been changed to "or 'physics' for short" as suggested

L24: 'Yano and Plant 2015' has been corrected to 'Plant and Yano 2015' as suggested

L24-25: We believe that references are still a metric used in evaluating science, so reviews cannot replace original work. Notwithstanding, the editor is right and we have added the references to Zhang *et al.* 2020, Jiang *et al.* 2020 as suggested (L25–26).

L39: 'dissipations' has been modified to 'dissipation' everywhere as suggested

L40, 'A classical work by Chang (1977) makes this point by invoking surface friction as a mechanism to slow down the propagation speed ...': Indeed, this is an important one of the earliest theoretical papers on the MJO, and also a good example to demonstrate how the role of friction has been considered to be important in the MJO dynamics. The main line of research on "convectively coupled waves" is already discussed in an earlier part (L24–26). Note that the purpose here is to establish the potential importance of friction in context of historical studies of MJO.

L45, 'When physical forcings are switched off from a model, an alternative mechanism for generating MJOs must also be considered.': This sentence was "puzzling" due to its unfortunate incompleteness. The sentence 'A shortcoming of the free–wave theory of the MJO is that it does not explain by itself an MJO initiation.' is added in revision (L47).

L49: The reference to Hall *et al.* (2016) is added in revision (L54–55), but as a reference for suggesting an importance of lateral forcing in simulating a MJO under an equatorial channel configuration. Note that this paper itself does not investigate a role of higher–latitude Rossby waves in the MJO dynamics in any explicit manner.

L57: The second question here has been modified in revision to "To what extent can the simulated MJO be interpreted in terms of free Rossby–wave dynamics?" In this context, the question of interactions of the MJO with higher–latitude Rossby waves also naturally arises as remarked over the paragraph of L70–75.

Table 1, 2nd row: The original description 'OFF selected or total physical tendency for the momentum (due to shallow and deep convection and the vertical eddy diffusion)' was rather confusing. However, change it to 'OFF selected for total physical tendency' does

not solve the problem, because tendencies are turned off either in selective manner or in totality. In revision, it is changed to 'OFF selected physical tendencies for the momentum (*e.g.*, shallow and deep convection, vertical eddy diffusion)'.

L67: As already responded at L57, the question 2) has been reformulated in revision.

L90, 'and it is likely also model specific': This non-essential remark has been removed in revision.

L103: Following the suggestions by the editor, the original lead sentences (L102–103) of this paragraph have been integrated into the paragraph L30–38 in the introduction. Though we considered to refer Virts and Wallace (2014), we found that this paper does not, in fact, use the stream function for their analysis.

L119: As suggested, this description (L118–120) has been integrated into the paragraph L30–38 in the introduction.

L122: This problematic sentence has simply been removed in revision.

L130: The phrase has been revised as "higher than the corresponding linear grid at the same spectral truncation".

L134: 'for the water substance' has been modified to 'for water substance' as suggested.

L136: 'the MJO in concern' has been modified to 'the MJO event considered here' by taking the suggestion

L140-141: 'how can we improve it' has been modified to 'how can we improve them', and 'in introduction' has been modified to 'in the introduction' as suggested.

L144-147: These two key questions here are simply whether it is possible to forecast those (two) key features found in the MJO evolution in these two forecasts cases. A phrase "on the other hand" (L155) has been added to suggest a relative separation of two questions, also in response to the Referee #2.

L175-180: The title of this subsection is re–named to "250-hPa Stream Function". Discussions about lower tropospheric rotational flow, etc are moved and merged to the beginning of Sec. 2.1.

L185: In revision, the section plots have been collocated based on the fields (OLR, 250-hPa stream function) and the periods (standard 20–day forecast, 40–day extended forecast, 20–day forecast from 1 Feb.): in all collocations, the first frame is the analysis field, thus it hopefully facilitates direct comparisons of forecasts with the analysis.

L195-205: The figure 2 (figure 6 in revision) has been revised following the suggestions.

L206-213: As the Editor correctly points out here, it is very difficult to interpret the plots of RMM as a measure of forecast skill. In fact, we believe that applicability of the RMM is much limited due to those inconsistencies (*e.g.*, mismatch of an initial point depending on forecast periods, etc). The point was already suggested in the previous version (L439–440: L448–449 in revision) by quoting Straub (2013), but it has been more explicitly stated in revision: "This design exactly becomes a key limitation of RMM" (L198–199); "However, afore–mentioned mismatching fundamentally limits the applicability of the RMM analysis in the following"(L215). Note that the RMM coordinates (patterns to project) are fixed in all the analyses. The real problem is in applying the RMM index, originally introduced to characterize the MJO in a long data set, to a short forecast run: the definition of temporal anomaly, to be projected, changes depending on the period considered.

L224, 'the model simulates those interactions features rather well' : To discuss forecast improvements by modified physics, the performance of a standard forecast (control runs) must first be established. In this case, all the forecasts considered simulate the interactions between the MJO and higher–latitude Rossby waves rather well, but without finding any noticeable improvements by modified physics.

L269, 'apparently in support of a free nonlinear-wave theory': This phrase has been removed as well as other similar phrases that may confuse the reader.

L402, 'failure of emulating the free dynamics ... maintaining a realistic mean state as detailed further below': A reference to Ma and Kuang (2016) has been added (L409) in revision. In the above sentence, "mean" has also been replaced by "background" in revision for consistency of the terminology. Note a subtle difference between "mean" (climatology) and "background" as discussed in revision (L425–428).

L421: Though conclusions regarding initial conditions, discussed here, are based only on two different initial conditions, we believe it worthwhile to mention, because, as already stated in the text, the finding is consistent with that of Kim *et al.* (2016), thus more likely to be generally the case than otherwise.

L441, 'The present study has also elucidated active interactions of MJOs with higher-latitude Rossby wave activities.': We believe that this statement itself stands with the present study, because the word "to elucidate" does not necessarily suggest any "systematic conclusions", which are missing as the Editor points out. The discussion has been much shortened in revision by removing remarks concerning 'westward dispersion of the original dipole' as well as 'Overall, forecasts of these interactions are found to be rather robust . . .'.

**Respond to the Referee #2**

Overall opinion

As the Editor summarises, the present Referee thinks the paper is clearer, but that it is still difficult to find the important points and to find any connection between the scientific questions posed and the results obtained. For this reason, the Referee suggests rejection.

We are glad to know that the present Referee finds significant improvements in the manuscript by stating "scientific questions become clearer, and the explanations become more detailed."

However, the Referee also remarks that "the main results are not sufficient to support two scientific questions". Indeed this negative result was clearly stated in the beginning of the final section in the last version of the manuscript (L393–396 in revision): "the present study does not support the proposed free nonlinear Rossby–wave theory in any consistent manner. Details on the forecast behaviour based on the choice of physical configurations of the model have been carefully documented to record the unexpected but nevertheless important impact on MJO forecast skill."

The "main points" were also already clearly stated in the previous version of the abstract: "A change of the forecast performance rather sensitively depends on the type of friction turned off. The behaviour is in contrast to many theoretical studies based on a rather simple Rayleigh–friction formulation under a dichotomy of with or without. By reporting the details of those physical sensitivities on the MJO forecast, the present study suggests a need for theoretical investigations that much more closely follow the actual operational formulations of physics. An important lesson to learn from the study is an inherent difficulty to emulate a free dynamics with an operational forecast model. The study also exposes the importance of other physical processes than convection for simulating the MJO in global forecast models."

To address the difficulties and sharpen those important, main points better, the above has been modified in revision to (L6–11):

"Contrary to the original motivation, emulating a free dynamics with an operational forecast model turns out to be rather difficult, because forecast performance sensitively depends on the specific type of friction turned off. The result suggests a need for theoretical investigations that much more closely follow the actual formulations of model physics: a naive approach with a dichotomy of with or without friction simply fails to elucidate the rich behaviour of complex operational models. The paper further exposes the importance of physical processes other than convection for simulating the MJO in global forecast models."

The discussion section has been revised and a new paragraph has been added (L397–409: see also L88–94 in the introduction).

Especially, the last two sentences of the new paragraph emphasise: "Though we are short of making any definite conclusions from our sensitivity study of the MJO on the momentum dissipation processes, the study suggests a critical importance of examining the physical sensitivities of a phenomenon with much detail rather than simply switching off the entire physical mechanism as has been done in past sensitivity studies." We believe that this is an important message that needs to be widely appreciated in the community.

Please also see the final paragraphs in L458–465.

This together with the re-structuring of the text following the Editor's suggestions, directly address the main objection of the present Referee that "it is hard to figure out the main points": we believe that this is now resolved in the present version of the manuscript.

Major comments

1. L143-147 "From a dynamical point of view, this is before the anticyclonic activity begins to develop over the Indian Ocean (Fig. 1(d)). Thus the key forecast question is whether the model can predict the onset of this activity". The Referee points out that these sentences are "not logically connected" with the previous sentences. This is correct, because we are making two independent statements. For clarity, in revision, we have added "On the other hand" in the beginning of the second pair of sentences (L155).

2. L279-281: Here, we suggest merely from a morphological basis that "the emission of the Rossby wave energy from west during 22-28 January is suggested as a major source for initiating the anticyclonic signal associated with the MJO". The word "suggested" was probably too strong, which has been replaced by "speculated" in revision (L281). However, yes, we speculate that the Rossby–wave energy is the source.

3. L316-317: Here, "additive" means that we can reproduce the same result of a run turning–off two processes both A and B by adding changes form two runs tuning off processes A and B, separately. Our result shows that it is clearly not the case. Yes, one of the possible consequences is that longer runs will lead to different results.

4. L297: The sentence in concern has been removed in revision, also by following a general suggestion by the Editor (his comment on L269).

5. L337-343: The word "significant" has been removed in revision. The role of the Rossby wave in this process is merely our speculation, thus we have added the phrase "we speculate that" in revision (L338).

6. Second scientific question and Figure 9: As the present Referee correctly suggests, the idea "that the Rossby-wave train is really important in this event" is not substantiated in any concrete manner by the present study, and we do not make any definite conclusion. Discussions concerning Fig. 9 merely focuses on morphological aspects of simulations, thus the Referee's suspects and objections about the presentation do not actually apply. [The second question itself has been re–phrased by following a suggestion of the Editor in revision.]

Note that NQ, QF, and Ma experiments are important, because we base the experimental setups on the hypothesis in the introduction that by realising a state closer to the free dynamics, interactions between the MJO and Rossby waves would also be enhanced. In this respect, sensitivities identified in Fig. 9 are rather intriguing, although we are short of making any definite conclusions. [Longitude–latitude plots were extensively examined, but we found it hard to choose selective snap shots to make a point: simply there are too many things going on in a field.]

Minor comments:

1. L5:"but hardly in any additive manner": This phrase has been modified to "but without any systematic tendency"

2. L32: The paragraph of L30–38 has been substantially expanded in revision, also by following the suggestion of the Editor.

3. L73 & L197, 4. L75 & L184: Please note that the present manuscript follows the British spelling rather than American, to which the present Referee appears to be more familiar with.

5. Figs. 1, 4, and 5: Please note that periods are not the same for all the plots, thus depending on extents of the periods, the $y$–axis also appears differently for obvious reasons. An aesthetic aspect of this problem has been removed in revision by re–collocating all the plots by the experiment periods and the physical fields in concern.

60. L266 : Probably, the phrase "this overall aspect" was obscure. It has been changed to "the overall aspect" in revision (L267). Of course, it means the overall aspect on the sensitivity on the choice of momentum dissipation terms, as clearly remarked in the proceeding sentence.

7. Page 26: The Caption for Figure. 6 was indicated as "Figure. 5" by editorial mistake in the previous version. The mistake has been corrected in revision, as now seen as Figure 3.

---

## Author Response (AR3)

**Response to the Editor**

We would like to thank the Editor for his second review, and respond concisely and brief as requested:

Crueger and Stevens (2015, JAMES) has been added to citation (L92). However, here also only a single "whole" process (cloud–radiative feedback) is turned off in this study.

An outline of the paper has been added at the end of introduction (L108–109)

The title of Sec. 2 has been modified into: Model, Forecast, Cases and Analysis Procedure.

Sec. 2.2 has been put before Sec. 2.1.

A reference to Sec. 2.1 (previous Sec.2.2) has been added where IFA is introduced as a model for analyses (L38)

To aid the interpretation of Fig. 1, the phrase "as identified as negative signals (in blue) stretching from the upper–left to the lower–right" has been added (L140–141).

Fig. 1 caption modified

L146, 'as expected from the nonlinear free-Rossby wave theory': a reference to Wang *et al.* (2019) has been added, in which further discussion are found about the Gill solution.

L144 (L150), "MJO events" → "MJO event"

L440–448: we agree with the statement 'better forecast for the wrong reasons'

General comment: To be discussed further, but we note that there are 70 Million matches when one enters in Google "the merits of single case studies", and notable attributes are given "as empirically-rich, context-specific, holistic accounts that they have to offer, and their contribution to theory-building". Notwithstanding that statistical significance cannot be claimed, our aim is the former.

All the other modifications have been made as suggested.